# Integrated control of transporter endocytosis and recycling by the arrestin-related protein Rod1 and the ubiquitin ligase Rsp5

Michel Becuwe[†], Sébastien Léon*

Department of Cell Biology, Institut Jacques Monod, Université Paris-Diderot, CNRS, Paris, France

**Abstract** After endocytosis, membrane proteins can recycle to the cell membrane or be degraded in lysosomes. Cargo ubiquitylation favors their lysosomal targeting and can be regulated by external signals, but the mechanism is ill-defined. Here, we studied the post-endocytic trafficking of Jen1, a yeast monocarboxylate transporter, using microfluidics-assisted live-cell imaging. We show that the ubiquitin ligase Rsp5 and the glucose-regulated arrestin-related trafficking adaptors (ART) protein Rod1, involved in the glucose-induced internalization of Jen1, are also required for the post-endocytic sorting of Jen1 to the yeast lysosome. This new step takes place at the trans-Golgi network (TGN), where Rod1 localizes dynamically upon triggering endocytosis. Indeed, transporter trafficking to the TGN after internalization is required for their degradation. Glucose removal promotes Rod1 relocalization to the cytosol and Jen1 deubiquitylation, allowing transporter recycling when the signal is only transient. Therefore, nutrient availability regulates transporter fate through the localization of the ART/Rsp5 ubiquitylation complex at the TGN.

*For correspondence: leon.sebastien@ijm.univ-paris-diderot.fr

Present address: †Department of Genetics and Complex Diseases, Harvard School of Public Health, Boston, United States

Competing interests: The authors declare that no competing interests exist.

## Introduction

Cells remodel the protein composition of the plasma membrane, such as receptors, transporters or channels, in response to external cues. Endocytosis is a major component of this adaptation. The conjugation of ubiquitin to membrane proteins ('cargoes') acts as a major determinant of their intracellular sorting in the endocytic pathway. Ubiquitin can notably act as a signal to promote cargo endocytosis from the plasma membrane (reviewed in *Dupré et al., 2004*). This is particularly true in yeast, and studies have shown the critical involvement of Rsp5, a conserved ubiquitin ligase of the Nedd4 family, in cargo ubiquitylation at the plasma membrane upon triggering endocytosis (reviewed in *MacGurn et al., 2012*).

The regulation of yeast nutrient transporters by nutritional signals in the environment has been used as a paradigm to dissect the mechanisms of signal-induced ubiquitylation and endocytosis (*Lauwers et al., 2010*). Indeed, the nutritional status of the medium, such as the absence/excess of a given substrate, regulates the dynamics of the cognate transporters at the plasma membrane through their regulated ubiquitylation and endocytosis. Recent work has notably contributed to our understanding of the mechanism responsible for the signal-induced ubiquitylation of transporters. Indeed, at the plasma membrane, the ubiquitin ligase Rsp5 ubiquitylates endocytic cargoes and is assisted in its function by adaptor proteins of the ART family (Arrestin-Related Trafficking adaptors), that confer to Rsp5 the ability to ubiquitylate numerous transporters in response to various external inputs (*Lin et al., 2008*; *Nikko et al., 2008*; *Nikko and Pelham, 2009*; *Hatakeyama et al., 2010*; *O'Donnell et al., 2010*, *2013*; *Karachaliou et al., 2013*). ART proteins are both targets of nutrient signaling pathways and actors of transporter ubiquitylation, and as such act as relay molecules allowing to coordinate transporter endocytosis with the nutrient status of the cell (*MacGurn et al., 2011*; *Merhi and André, 2012*;

**eLife digest** The plasma membrane that surrounds cells contains many different proteins that perform tasks such as detecting signals sent to the cell, and transporting molecules into or out of the cell. To adapt to changing conditions, cells remodel their membrane to change how much of each type of protein is present. A process called endocytosis—where part of the plasma membrane and the proteins it contains buds off into the cell—plays an important role in this remodeling.

The fate of a membrane protein after endocytosis can depend on whether a protein 'tag' called ubiquitin has been added to it. Ubiquitin-marked proteins bud off into the cell and are then sent to cell structures called lysosomes to be degraded, whereas unmarked proteins are recycled back to the plasma membrane.

Yeast cell membranes contain a protein called Jen1 that transports certain molecules, including one called lactate that can be used as fuel for growth. However, glucose is a preferred nutrient for yeast, so when glucose is available, another protein called Rod1 becomes activated and promotes the addition of ubiquitin to Jen1, and hence its degradation. This means that the cells can no longer use lactate as a source of energy. However, it was not known where in the cell the Rod1 protein does this.

Becuwe and Léon labeled proteins involved in endocytosis with fluorescent tags and used microscopy to observe their fate in live yeast cells exposed to glucose. This revealed two roles for Rod1. At the plasma membrane, Rod1 helps Jen1 to be taken into the cell in the early stages of endocytosis. But unexpectedly, Rod1 is also found at a cellular structure called the trans-Golgi network, small membrane sacs that are typically responsible for packaging proteins so they can be transported to a new destination, in particular the plasma membrane. This suggests that Rod1 can also act at this location in the cell.

When the proteins responsible for maintaining transport to the trans-Golgi network are inhibited, Jen1 is no longer degraded, even when glucose is present; instead, Jen1 is recycled back to the plasma membrane. Becuwe and Léon therefore propose that a second level of control of the degradation of plasma membrane proteins occurs in the trans-Golgi network, and so this compartment has an essential role in sorting proteins for degradation or recycling.

The group of proteins that Rod1 belongs to, named arrestins, has been suggested to play important roles in several diseases, including diabetes and cancer. As many of the features of the endocytic pathway are conserved in a broad range of species, arrestins may also be important for controlling the fate of membrane proteins at multiple places in mammalian cells. However, further work is required to confirm this.

*Becuwe et al., 2012b*). Interestingly, the human arrestin-related protein TXNIP was recently reported to regulate glucose influx in cells by promoting the endocytosis of the glucose transporter GLUT1 in an AMP-activated protein kinase (AMPK)-dependent manner, illustrating the conservation of arrestin-related proteins function in endocytosis (*Wu et al., 2013*).

In mammalian cells, cargo ubiquitylation often occurs at the plasma membrane, and in some cases, this is essential for internalization, as demonstrated for the sodium channel ENaC (*Zhou et al., 2007*) or the Major Histocompatibility Complex (MHC) class I (*Hewitt et al., 2002*). However, this is not a general rule, because affecting the ubiquitylation of the G-protein-coupled receptors β2-AR and CXCR4 (*Shenoy et al., 2001*; *Marchese and Benovic, 2001* #885), or the receptor tyrosine kinases EGFR and FGFR-1 (*Huang et al., 2006*; *Haugsten et al., 2008*) does not abrogate their internalization. Instead, cargo ubiquitylation becomes critical later in the endocytic pathway, to promote cargo sorting towards late endosomes and their subsequent degradation (see for instance *Miranda et al., 2007*; *Kabra et al., 2008*). At endosomes, ubiquitinated cargoes are captured and sorted by ESCRT proteins (Endosomal Sorting Complex Required For Transport) into the lumen of the endosome through a process known as multivesicular body (MVB) biogenesis (*MacGurn et al., 2012*). Loss of cargo ubiquitylation reportedly leads to their recycling to the cell surface (*Shenoy et al., 2001*; *Huang et al., 2006*; *Haugsten et al., 2008*; *Eden et al., 2012*). Therefore, a major function of ubiquitin conjugation to endocytic cargoes is to regulate their post-endocytic fate.

Because ubiquitylation is a reversible modification, the competition between ubiquitylation and deubiquitylation at endosomes can decide the fate of many of the endocytosed cargoes, as only

proteins that are stably ubiquitylated will be ultimately degraded. In mammalian cells, several deubiquitylating enzymes were indeed shown to control the recycling of endosomal cargoes such as EGFR (*McCullough et al., 2004*; *Mizuno et al., 2005*; *Row et al., 2006*), the Cystic Fibrosis Transmembrane Regulator (CFTR) (*Bomberger et al., 2009*), ENaC (*Butterworth et al., 2007*) or the β2-adrenergic receptor (β2-AR) (*Berthouze et al., 2009*) (reviewed in *Clague et al., 2012*). However, the mechanisms by which external signals are integrated and transduced to the ubiquitylation machinery in order to regulate cargo trafficking are unknown.

We recently reported that glucose, which triggers the endocytosis of various yeast carbon sources transporters (reviewed in *Horak, 2013*), promotes the successive dephosphorylation and ubiquitylation of the ART protein Rod1 (also known as Art4), leading to its activation (*Becuwe et al., 2012b*). In turn, activated Rod1 promotes the glucose-induced ubiquitylation and degradation of Jen1, a monocarboxylate transporter. However, the place where Rod1 regulates endocytosis is unknown. Whereas ART proteins are considered to promote transporter internalization (*Becuwe et al., 2012a*), there are conflicting reports regarding the step at which they control intracellular trafficking (*Helliwell et al., 2001*; *Soetens et al., 2001*; *Merhi and André, 2012*). Moreover, ART proteins were observed at various subcellular locations, both in yeast (*Lin et al., 2008*; *O'Donnell et al., 2010*) and mammalian cells (*Vina-Vilaseca et al., 2011*; *Han et al., 2013*) suggesting other roles beyond the regulation of cargo internalization. Furthermore, it was reported that akin to the situation in mammalian cells, the yeast iron transporter complex Fet3/Ftr1 is internalized independently of its ubiquitylation, although the latter is required for its vacuolar targeting. This suggests a control of transporter degradation through a post-endocytic ubiquitylation event, for which the molecular actors are not known (*Strochlic et al., 2008*).

Here, through the use of microfluidics-assisted live-cell imaging, we studied the place of action of Rod1 during endocytosis and report a dual function of Rod1 in transporter sorting at two successive locations in the cell, that is, at the plasma membrane to control transporter internalization, and at a post-endocytic compartment to promote vacuolar sorting and degradation.

## Results

### Rod1 contributes to transporter internalization at the plasma membrane

As a first approach to identify the place of action of Rod1 in endocytosis, we set out to precisely document the endocytosis defect displayed by the *rod1Δ* mutant. For this purpose, we monitored transporter trafficking in wild type (WT) and *rod1Δ* cells using two different cargoes, namely the glycerol/proton symporter Stl1, and the monocarboxylate transporter Jen1. Stl1 is expressed when cells are grown in glycerol/lactate medium, and endocytosed in response to glucose (*Ferreira et al., 2005*). The glucose-induced endocytosis and degradation of Stl1 required Rod1, establishing Stl1 as a new Rod1-regulated cargo (*Figure 1A,B*), in addition to the previously described hexose transporter Hxt6 (*Nikko and Pelham, 2009*) and Jen1 (*Becuwe et al., 2012b*). We then performed time-lapse microscopy using a microfluidics device to precisely monitor Stl1-GFP localization immediately after glucose addition in WT and *rod1Δ* cells. Both strains were co-injected into the microfluidics chamber and observed simultaneously. Their identification was made possible by a pre-staining of the *rod1Δ* cells with the vital dye CMAC. Whereas Stl1-GFP was internalized within 5 min after glucose addition in WT cells, it remained stably associated to the plasma membrane in the *rod1Δ* mutant and was not internalized even 30 min after glucose treatment (*Figure 1C*, *Video 1*). This is in agreement with a canonical role of Rod1 in transporter internalization at the plasma membrane.

### Rod1 is involved in the post-endocytic sorting of Jen1 to the vacuole

Then, we monitored the trafficking of the monocarboxylate transporter Jen1-GFP in *rod1Δ* cells after glucose addition. We observed that, in sharp contrast with the result obtained for Stl1 (see *Figure 1C*), glucose triggered the transient localization of Jen1 to cytoplasmic puncta (*Figure 1D*, *Video 2*). The appearance of these puncta was strongly affected by latrunculin A treatment, which disrupts the actin cytoskeleton and abolishes endocytosis, indicative of their endocytic origin (*Figure 1E*). This showed that Jen1 was still internalized in the *rod1Δ* mutant. To evaluate the contribution of Rod1 in Jen1 internalization, we then quantitatively compared Jen1 trafficking in both WT and *rod1Δ* cells using microfluidics (*Figure 1F*, *Video 3*). First, we observed that the appearance of Jen1-positive vesicles was delayed in the *rod1Δ* mutant as compared to the wild type (*Figure 1G*). This clearly showed

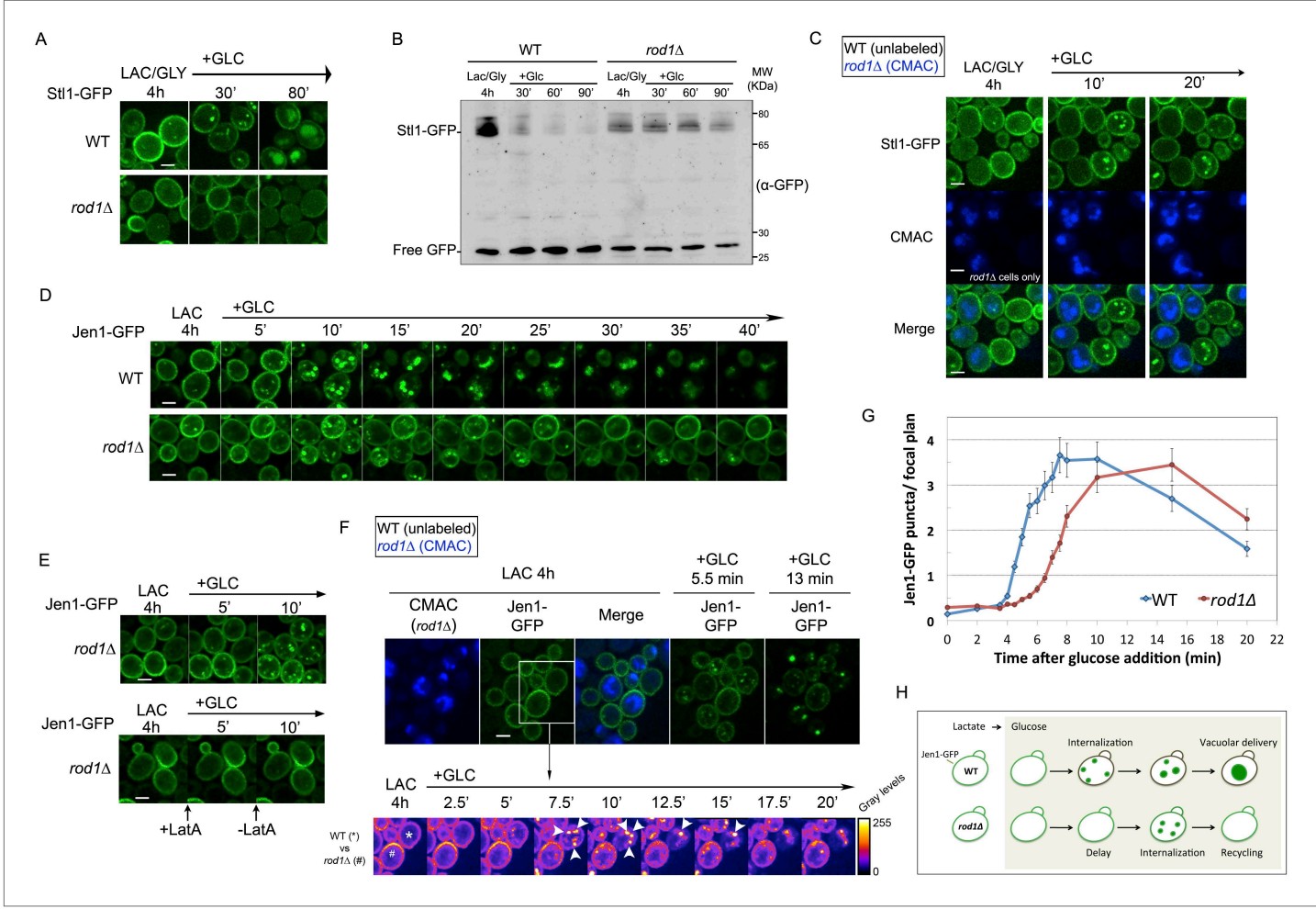

**Figure 1**. Dual function of Rod1 in transporter internalization and post-endocytic sorting. (**A**) Rod1 is required for the glucose-induced endocytosis of Stl1, the glycerol/proton symporter, from the plasma membrane to the vacuole. WT (ySL1146) and *rod1Δ* (ySL1153) cells were grown in lactate/glycerol medium to induce Stl1-GFP expression and targeting to the plasma membrane. Cells were treated with glucose for the indicated times and imaged for Stl1-GFP localization. Scale bar = 2.5 μm. (**B**) Stl1-GFP degradation in response to glucose requires Rod1. WT (ySL1146) and *rod1Δ* (ySL1153) cells expressing Stl1-GFP were grown as in **A**. Crude extracts were prepared at the indicated times and were immunoblotted with anti-GFP antibodies. (**C**) Rod1 is required for Stl1 internalization in response to glucose. WT (ySL1146) and *rod1Δ* (ySL1153) cells were grown in lactate/glycerol medium to induce Stl1-GFP expression and targeting to the plasma membrane. *rod1Δ* cells were then labeled with CMAC and were co-injected with WT cells into the microfluidics device in lactate/glycerol medium, before glucose was added. Images taken at 10 and 20 min after glucose addition are shown. Scale bar = 2.5 μm. See also *Video 1*. (**D**) Jen1-GFP is internalized upon glucose treatment even in the absence of Rod1. Lactate-grown WT (ySL1150) and *rod1Δ* (ySL743) cells expressing Jen1-GFP were injected into a microfluidics device in lactate medium. Cells were imaged over time after glucose addition. Scale bar = 2.5 μm. (**E**) The appearance of Jen1-GFP-positive puncta in the *rod1Δ* mutant is inhibited by latrunculin A (LatA). Left panel, *rod1Δ* (ySL743) cells expressing Jen1-GFP were grown on lactate medium and injected into the microfluidics device in lactate medium, before glucose was added. Right panel, glucose and LatA were simultaneously added. After 5 min, LatA was removed and cells were fueled only with glucose medium. Scale bar = 2.5 μm. (**F**) *rod1Δ* cells display a kinetic delay in Jen1 internalization. Top, WT (ySL1150) and *rod1Δ* (ySL743) cells expressing Jen1-GFP were grown on lactate medium. The *rod1Δ* cells were then labeled with CMAC and were co-injected with WT cells into the microfluidics device in lactate medium, before glucose was added. Images taken at 5 and 13 min after glucose addition are shown. Bottom, images representative of WT and *rod1Δ* cells are shown at various times and are shown in false colors to visualize Jen1 fluorescence intensity. Arrowheads indicate strongly fluorescent vesicles, presumably late endosomes, which do not appear in the *rod1Δ* mutant. Scale bar = 2.5 μm. See also *Video 3*. (**G**) Quantification of the experiment shown in **F**. The mean number (±SEM) of vesicles in a focal plane for each strain (30 cells/strain, *n* = 3) was plotted as a function of time. (**H**) Graphical representation of the phenotype observed in *rod1Δ* cells. A fraction of Jen1 is internalized but recycles to the cell membrane.

that in the absence of Rod1, Jen1 internalization still occurred but was less efficient, which was also supported by the persistence of a Jen1-GFP pool at the plasma membrane in the *rod1Δ* strain. A second observation was that whereas Jen1-GFP was targeted into larger and brighter structures (likely to

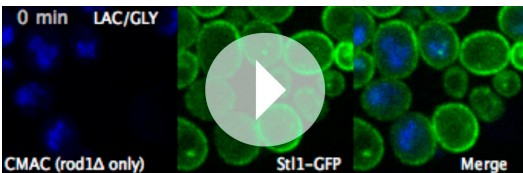

**Video 1**. Rod1 is required for the glucose-induced internalization of the glycerol/proton symporter Stl1. WT and *rod1Δ* (CMAC-positive) cells expressing Stl1-GFP were grown in lactate/glycerol medium and simultaneously observed for 20 min after glucose addition. See also *Figure 1C*.

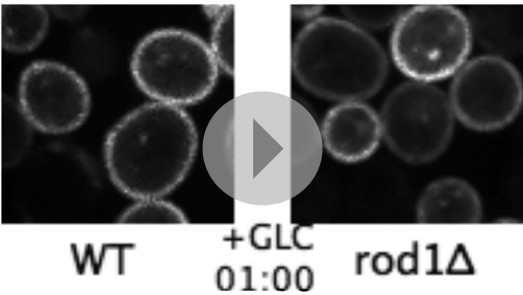

**Video 2**. Jen1-GFP is internalized upon glucose treatment even in the absence of Rod1. WT cells (left) and in *rod1Δ* cells (right) expressing Jen1-GFP were grown in lactate medium and observed for 45 min after glucose addition. See also *Figure 1D*.

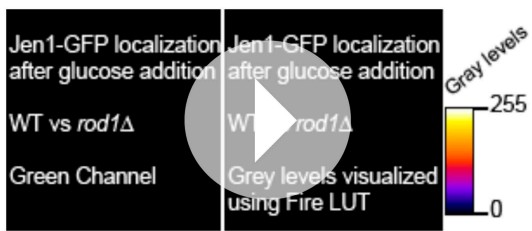

**Video 3**. rod1Δ cells display a kinetic delay in Jen1 internalization. WT and rod1Δ (CMAC positive) cells expressing Jen1-GFP were visualized simultaneously during 20 min after glucose addition (left). Images of the same video were treated in ImageJ using the 'Fire' lookup table (LUT) to visualize pixel intensity (right). See also *Figure 1F*.

be late endosomes) at later time points in the WT, it did not reach this compartment in the *rod1Δ* mutant (*Figure 1F*, *Video 3*) but rather re-localized to the plasma membrane, as described previously (*Becuwe et al., 2012b*) (see also *Figure 1D* and *Video 2*). Because *JEN1* expression is repressed by glucose (*Bojunga and Entian, 1999*), this plasma membrane-localized pool did not originate from de novo Jen1 synthesis, but rather from the recycling of internalized Jen1 back to the cell surface. This result strongly suggested a role for Rod1 in the post-endocytic targeting of Jen1 to the vacuole, in addition to its function at the plasma membrane (*Figure 1H*).

Because we previously showed that Rod1 is required for an efficient glucose-induced ubiquitylation of Jen1 (*Becuwe et al., 2012b*), the observation that Jen1 can be partially internalized in the absence of Rod1 led us to investigate whether this endocytosis still relies on Jen1 ubiquitylation. Indeed, the remnant internalization of Jen1 observed in the *rod1Δ* mutant still required its ubiquitylation, as demonstrated through the use of a non-ubiquitylatable Jen1 mutant in which all cytosolic lysine residues have been mutated into arginine residues (Jen1-KR-GFP, *Figure 2A*). As expected, these mutations abolished Jen1 ubiquitylation in response to glucose (*Figure 2B*), but this construct was still functional, as judged by its ability to transport selenite, that can be used as a readout for Jen1 activity (*Figure 2—figure supplement 1*) (*McDermott et al., 2010*). The visualization of the subcellular localization of Jen1-KR-GFP showed that it was indeed targeted to the plasma membrane, although it displayed a small delay in secretion as observed by its transient accumulation at the ER (*Figure 2C*). More importantly, the endocytosis of Jen1-KR-GFP in response to glucose was abolished (*Figure 2C*). Since Jen1 was partially endocytosed in the absence of Rod1 (see *Figure 1D,F,G*), we conclude that Jen1 can be ubiquitylated at the plasma membrane independently of Rod1, at least when Rod1 is absent. Consequently, it is likely that there are additional ubiquitylation systems besides Rod1 at the plasma membrane, which can promote Jen1 internalization and that remain to be identified. Of note, Jen1 was still internalized in the multiple *9-arrestin* mutant (*Figure 2D*), which lacks Rod1 and eight other arrestin-related proteins (*Nikko and Pelham, 2009*). Altogether, these results show that the function of Rod1 at the plasma membrane is either compensated or is redundant with other adaptors, but that this is not the case regarding its post-endocytic function in Jen1 trafficking.

## Rod1 dynamically localizes to the trans-Golgi network in response to glucose

To identify the additional compartment at which Rod1 controls the post-endocytic trafficking of Jen1, we next investigated the subcellular localization of Rod1. Rod1-GFP appeared as a diffuse cytosolic

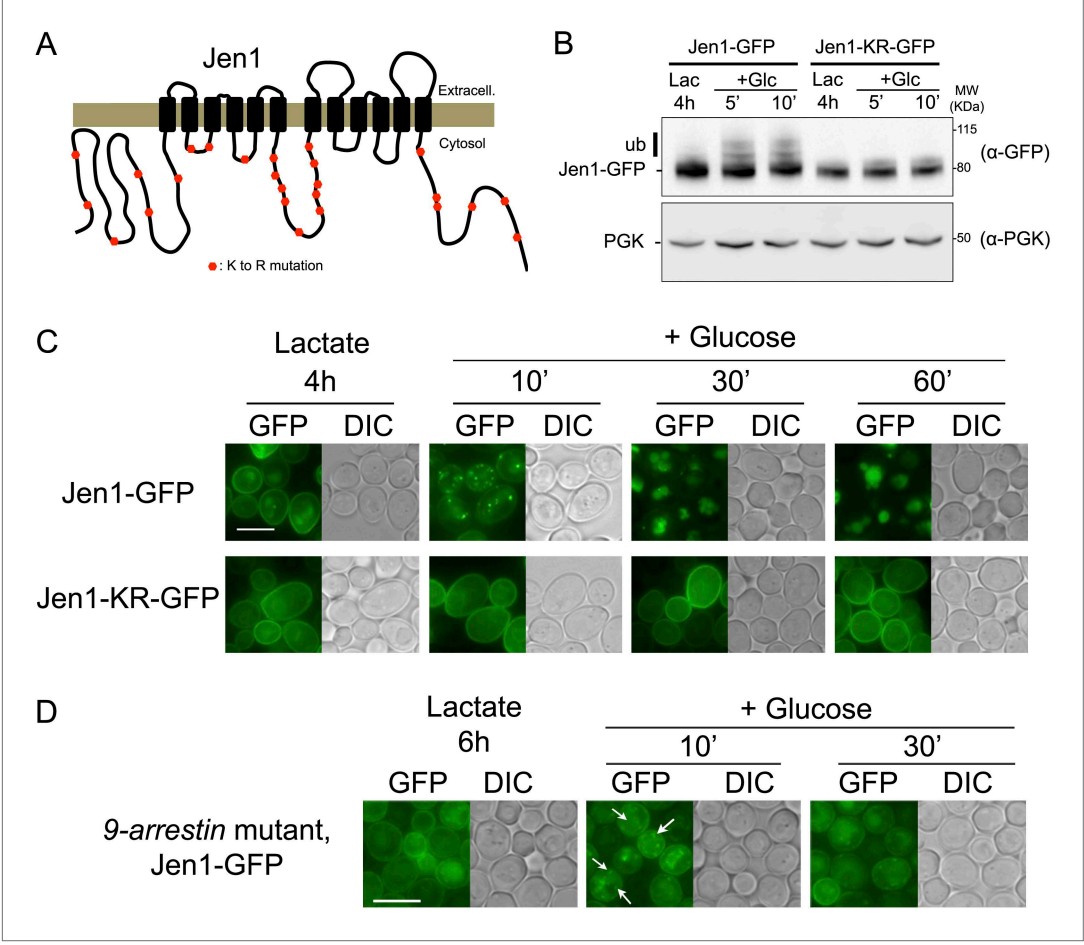

**Figure 2**. Jen1 ubiquitylation is required for its glucose-induced endocytosis. (**A**) Schematic of the lysine-to-arginine mutations introduced in the cytosolic loops of Jen1 to generate the Jen1-KR construct. (**B**) Jen1-KR-GFP is not ubiquitylated in response to glucose. WT cells carrying a plasmid-encoded Jen1-GFP (pSL161) or the same plasmid bearing the KR mutations (pSL163) were grown on lactate medium and glucose was added for the indicated times. Crude extracts were immunoblotted with the indicated antibodies. Yeast PGK (phosphoglycerate kinase) is used as a loading control. Appearance of the glucose-induced higher molecular weight species of Jen1, which were previously shown to correspond to Jen1 ubiquitylated adducts (**Paiva et al., 2009**; **Becuwe et al., 2012b**), do not appear upon mutations of Jen1 lysines. (**C**) Jen1-KR-GFP is not internalized in response to glucose. WT cells carrying a plasmid-encoded Jen1-GFP (pSL161) or the same plasmid bearing the KR mutations (pSL163) were grown on lactate medium, then glucose was added and cells were imaged at the indicated times. Upon glucose addition, Jen1-GFP is internalized into vesicles and then is targeted to the vacuole, but Jen1-KR-GFP remains stable at the plasma membrane. Note the partial endoplasmic reticulum (ER) labeling of Jen1-KR-GFP, showing that this construct displays a mild defect in ER exit to the secretory pathway. Scale bar = 5 µm. (**D**) Deletion of nine genes encoding arrestin-related proteins is not sufficient to abolish Jen1 internalization. The *9-arrestin* strain (strain EN60, a kind gift from Hugh Pelham: *art1Δ ecm21Δ aly2Δ rod1Δ art5Δ aly1Δ rog3Δ csr2Δ art10Δ*) (**Nikko and Pelham, 2009**) expressing Jen1-GFP tagged at its endogenous genomic locus (ySL1318) was grown in lactate medium. Glucose was then added to the medium and Jen1-GFP localization was monitored at the indicated times. Vesicles resulting from Jen1-GFP internalization can still be observed in this mutant after glucose addition (white arrows). Scale bar = 5 µm.
The following figure supplement is available for figure 2:

**Figure supplement 1**. Jen1-KR-GFP is a functional protein.

protein in lactate-grown cells that transiently localized to punctate structures upon glucose stimulation (*Figure 3A*, *Video 4*). This localization was due to the recruitment of pre-existing Rod1 to these puncta, because inhibition of translation with cycloheximide did not affect this localization (*Figure 3B*).

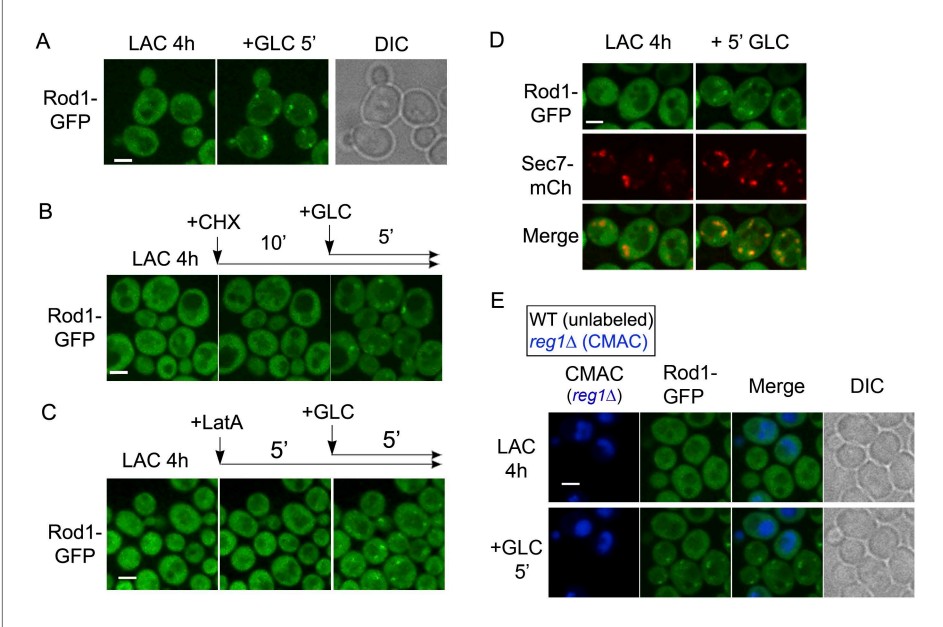

**Figure 3**. Rod1 is dynamically recruited to the trans-Golgi network when endocytosis is triggered. (**A**) Rod1-GFP re-localizes from the cytosol to punctate structures in response to glucose. Lactate-grown cells (ySL542) expressing Rod1-GFP were injected into a microfluidics device in lactate medium, and were then imaged over time after glucose addition. Scale bar = 2.5 µm. See also *Video 4*. (**B**) Inhibition of translation using cycloheximide (CHX) does not alter Rod1-GFP re-localization. Cells (ySL542) expressing Rod1-GFP were grown on lactate medium and injected into a microfluidics device in lactate medium. Cells were then treated with 100 µg/ml CHX for 10 min in lactate medium and imaged over time after addition of glucose in the presence of CHX. Scale bar = 2.5 µm. (**C**) Disruption of the actin cytoskeleton using latrunculin A (LatA) does not alter Rod1-GFP re-localization. Cells (ySL542) expressing Rod1-GFP were grown on lactate medium and injected into a microfluidics device in lactate medium. Cells were then treated with 0.2 mM LatA for 5 min in lactate medium, then imaged over time after addition of glucose in the presence of LatA. Scale bar = 2.5 µm. (**D**) Rod1 co-localizes with the trans-Golgi network marker, Sec7-mCherry, in response to glucose. Cells (ySL638) expressing both Rod1-GFP and Sec7-mCh were grown on lactate medium and injected into a microfluidics device in lactate medium. Cells were then imaged over time after glucose addition. Scale bar = 2.5 µm. See also *Video 5*. (**E**) Rod1 does not localize to the TGN in response to glucose in the mutant for the regulatory subunit of protein phosphatase 1, *reg1Δ*. WT (ySL542) and *reg1Δ* (ySL600) cells expressing Rod1-GFP were grown on lactate medium. The *reg1Δ* cells were then labeled with CMAC and were co-injected with WT cells into the microfluidics device in lactate medium, before glucose was added. Images taken before and after 5 min of glucose treatment are shown. Scale bar = 2.5 µm.

Also, these puncta did not originate from the plasma membrane by endocytosis because Rod1 behaved similarly when endocytosis was abolished by latrunculin A treatment (*Figure 3C*). Instead, these puncta co-localized with Sec7-mCh (*Figure 3D*, *Video 5*), a marker of the TGN (trans-Golgi network) (*Franzusoff et al., 1991*), indicating that Rod1 transiently localizes to the TGN in response to glucose. Deletion of *REG1*, encoding a subunit of protein phosphatase 1 (PP1) required for Rod1 activation (*Becuwe et al., 2012b*), abolished Rod1 targeting to the TGN, demonstrating that Rod1 localization to the TGN was a consequence of its glucose-induced activation (*Figure 3E*).

## Jen1 localizes to the TGN after endocytosis and fails to be targeted to the vacuole in the absence of Rod1

Our data showing a role for Rod1 in the post-endocytic sorting of Jen1, as well as the glucose-induced localization of Rod1 to the TGN, prompted us to delineate the compartments with which Jen1 associates during its endocytosis. We tested Jen1 co-localization with either the endosomal marker Vps17-mCh, a component of the recycling complex Retromer (*Seaman, 2004*; *Strochlic et al., 2008*) or the TGN marker Sec7-mCh, two proteins that did not co-localize with each other in wild-type cells as determined in a control experiment (*Figure 4—figure supplement 1*). Early after glucose treatment, the

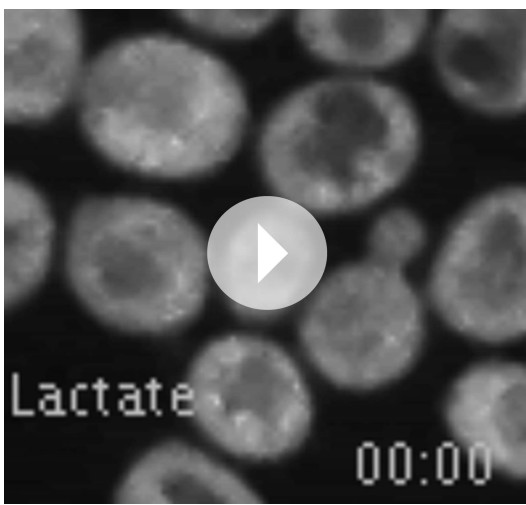

**Video 4**. Rod1-GFP relocalizes from the cytosol to punctate structures in response to glucose. WT cells expressing Rod1-GFP were grown in lactate medium and visualized for 60 min after glucose addition. See also **Figure 3A**.

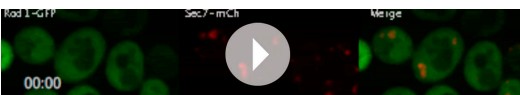

**Video 5**. Rod1 co-localizes with the trans-Golgi network marker, Sec7-mCherry, in response to glucose. WT cells expressing both Rod1-GFP and Sec7-mCh were grown during 4 hr in lactate medium and visualized during 45 min after glucose addition. Merge of Rod1-GFP fluorescence (left panel) and Sec7-mCh fluorescence (middle panel) is observed on the right panel. See also **Figure 3D**.

Jen1-GFP puncta co-localized nearly exclusively with Vps17-mCh, but this was followed by a partial co-localization with Sec7-mCh (**Figure 4A,C**, **Figure 4—figure supplement 2**, **Video 6**), showing that Jen1-GFP associates with the TGN after internalization. Although Jen1-GFP and Sec7-mCh sometimes appeared to be in adjacent structures, rather than completely overlapping, this was likely an artifact due to the fast mobility of TGN structures in yeast (**Kojima et al., 2012**), which led to a small shift between the image acquisition in each channel. Indeed, the simultaneous observation of Sec7-mCh and Jen1-GFP using a microscope equipped with two cameras showed a complete overlap of these signals (**Figure 4—figure supplement 3**). The same co-localization analysis was carried out in the *rod1Δ* mutant, and revealed that Jen1-GFP followed essentially the same route as in WT cells, although Jen1-GFP displayed a prolonged co-localization with Sec7-mCh (**Figure 4B,C**, **Figure 4—figure supplements 4, 5**, **Video 7**). This was not due to an indirect effect of *ROD1* deletion on organelle identity, because Vps17 and Sec7 were still observed in distinct compartments in the *rod1Δ* strain (**Figure 4—figure supplement 6**). Therefore, Jen1 not only traffics to the TGN after internalization before it reaches the vacuole in wild-type cells, but also transiently accumulates at this compartment in the absence of Rod1.

## Jen1 targeting to the vacuole after endocytosis involves a step at the TGN

To address whether the observed localization of Jen1 to the TGN is required for its vacuolar degradation, we probed Jen1 trafficking in mutants affected in retrograde, endosomes-to-TGN trafficking. We notably used a mutant *(vps52Δ)* of the Golgi-localized VFT (Vps Fifty-Three)/GARP (Golgi-associated retrograde protein) complex, involved in the fusion of vesicles of endosomal origin with the TGN (**Conibear and Stevens, 2000**; **Siniossoglou et al., 2000**; **Siniossoglou and Pelham, 2001**) (**Figure 4D**). In this strain, Jen1 was internalized but accumulated into cytoplasmic puncta and failed to reach the vacuole (**Figure 4E**). Similar results were obtained in other mutants affected in retrograde trafficking, such as mutants for the yeast Rab6 homologue, *ypt6Δ*, or for components of its GEF (guanine nucleotide exchange factor), *rgp1Δ* and *ric1Δ* (**Siniossoglou et al., 2000**) (**Figure 4E**). Moreover, Jen1 degradation was strongly affected in these retrograde mutants upon glucose treatment (**Figure 4F**). As expected, in the *ypt6Δ* mutant, Jen1 no longer co-localized with the TGN (**Figure 4G**). Therefore, these data reveal that endosome-to-TGN trafficking is critical for the vacuolar sorting of Jen1 after its internalization, suggesting that a step in the post-endocytic of Jen1 occurs at the TGN, prior to its delivery to the vacuole.

In order to identify the components involved in the vacuolar sorting of TGN-localized Jen1, we studied Jen1 trafficking in mutants of genes encoding the Golgi-localized clathrin adaptor proteins Gga1 and Gga2, involved in protein sorting from the TGN to endosomes (**Scott et al., 2004**) (see **Figure 4D**). As shown in **Figure 4H**, the absence of the Gga1/Gga2 proteins did not reduce Jen1 internalization, but partially impaired its vacuolar targeting, leading to Jen1 recycling at the plasma membrane. Of note, Jen1-GFP co-localized more strongly with the TGN marker, Sec7-mCh,

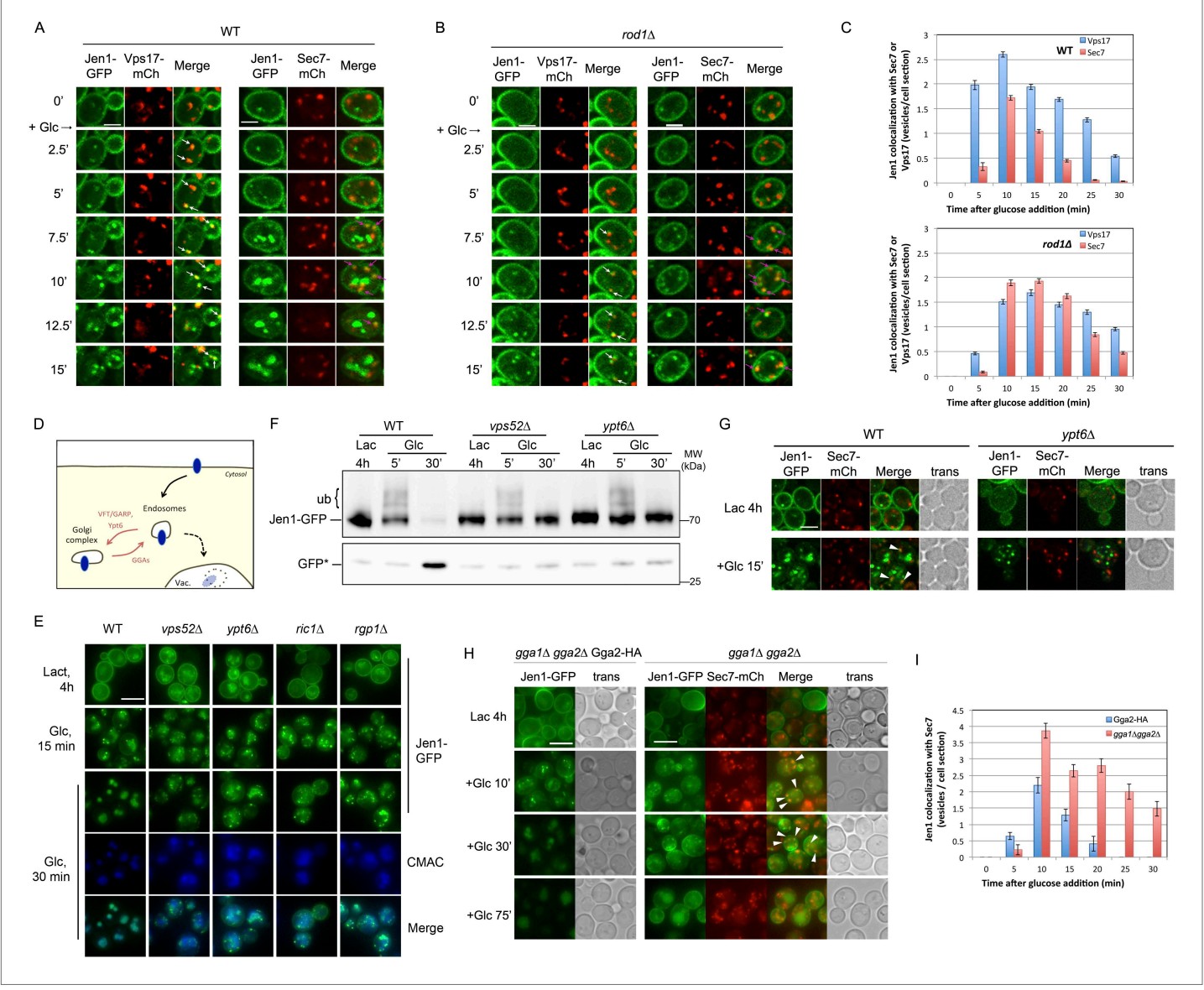

**Figure 4**. Jen1 transits through the TGN during its endocytosis and requires Rod1 for exit from the TGN to the vacuole. (**A**) Jen1 co-localizes with the TGN marker, Sec7-mCh, during its trafficking to the vacuole in wild-type cells. WT cells expressing Jen1-GFP and either Vps17-mCh (ySL1168), a marker of the early endosomal compartment, or Sec7-mCh (ySL1165) were grown on lactate medium. Of note, little or no co-localization was observed between Sec7-mCh and Vps17-GFP (*Figure 4—figure supplement 1*). The Sec7-mCh-expressing cells were labeled with CMAC and were co-injected with the Vps17-mCh-expressing cells into the microfluidics device in lactate medium, before glucose was added. Cells have been cropped to show only one Vps17-mCh expressing cell (left) or one Sec7-mCh expressing cell (right). The uncropped picture is displayed in *Figure 4—figure supplement 2*, and the full video in *Video 6*. Co-localization events between Jen1-GFP and either Vps17-mCh or Sec7-mCh are indicated with white arrows or pink arrows, respectively. Scale bar = 2.5 μm. Of note, the co-localization of another transporter, Dip5, with Sec7-mCh after its endocytosis was also observed (see *Figure 4—figure supplement 3*). (**B**) Jen1 also co-localizes with the TGN marker, Sec7-mCh, after internalization in the *rod1Δ* mutant. *rod1Δ* cells expressing Jen1-GFP and either Vps17-mCh (ySL1177), an endosomal marker, or Sec7-mCh (ySL1176) were grown on lactate medium. As in panel **A**, the Sec7-mCh-expressing cells were labeled with CMAC and were co-injected with Vps17-mCh-expressing cells into the microfluidics device in lactate medium, before glucose was added. Cells have been cropped to show only one Vps17-mCh expressing cell (left) or one Sec7-mCh expressing cell (right). The uncropped picture is displayed in *Figure 4—figure supplement 4*, and the full video in *Video 7*. Co-localization events between Jen1-GFP and either Vps17-mCh or Sec7-mCh are indicated with white arrows or pink arrows, respectively. Scale bar = 2.5 μm. (**C**) Quantification of the co-localization of Jen1-GFP puncta with either Vps17-mCh or Sec7-mCh puncta in WT (top) or *rod1Δ* (bottom) cells (20 cells, n = 3). Jen1 co-localizes successively with Vps17 and Sec7. Furthermore, Jen1 co-localizes more robustly with Sec7-mCh in the *rod1Δ* mutant. See also *Figure 4—figure supplement 5*. (**D**) Schematic showing the place of action of the VFT/GARP complex, Ypt6 and the GGA proteins in endosome-to-Golgi trafficking. (**E**) Deletions of *VPS52* (VFT/GARP complex), *YPT6* or genes encoding the Ypt6 GEF complex (*RGP1* and *RIC1*) abolish Jen1 trafficking to the vacuole. Lactate-grown

*Figure 4. Continued on next page*

*Figure 4. Continued*

WT (ySL1150), *vps52Δ* (ySL1369), *ypt6Δ* (ySL1175), *ric1Δ* (ySL1630) and *rgp1Δ* (ySL1631) cells expressing Jen1-GFP were imaged before or after the addition of glucose at the indicated time. At t = 30 min Glc treatment, CMAC staining was used to visualize the localization of the vacuole. Scale bar = 5 µm. (**F**) Deletions of *VPS52* or *YPT6* prevent the vacuolar degradation of Jen1-GFP. Crude extracts from lactate-grown WT (ySL1150), *vps52Δ* (ySL1369) and *ypt6Δ* (ySL1175) cells expressing Jen1-GFP were prepared at the indicated times before and after glucose addition, and were immunoblotted with anti-GFP antibodies to reveal the full-length Jen1-GFP and its degradation product (free GFP). (**G**) Deletion of *YPT6* abrogates Jen1 co-localization with Sec7-mCh. Lactate-grown WT cells (ySL1165) or *ypt6Δ* cells (ySL1526) expressing Jen1-GFP and Sec7-mCh were injected in a microfluidics device and imaged before (Lactate) and 15 min after glucose addition. Co-localization between Jen1-GFP and Sec7-mCh is indicated with arrowheads. (**H**) Deletions of *GGA1* and *GGA2*, encoding redundant Golgi-localized clathrin adaptor proteins, alter Jen1 trafficking to the vacuole. Strains *gga1Δgga2Δ* (ySL1307) or *gga1Δ gga2Δ* expressing Gga2-HA (ySL1308), used as a positive control, both expressing Jen1-GFP were grown on lactate medium, and imaged before and after the addition of glucose at the indicated times. The *gga1Δ gga2Δ* cells also express Sec7-mCherry, which allows evaluating of Jen1-GFP co-localization with the TGN (arrowheads). Scale bar = 5 µm. (**I**) Quantification of the co-localization of Jen1-GFP puncta with Sec7-mCh puncta in *gga1Δgga2Δ* (ySL1307) cells or *gga1Δ gga2Δ* cells expressing Gga2-HA (ySL1619) (20 cells, n = 3) over time after glucose addition. Deletion of the *GGA* genes leads to a transient accumulation of Jen1 at the TGN.

The following figure supplements are available for figure 4:

**Figure supplement 1**. Sec7 and Vps17 localize to distinct compartments.

**Figure supplement 2**. Jen1 traffics through the TGN in the course of its endocytosis in wild-type cells.

**Figure supplement 3**. Jen1-GFP and Sec7-mCh co-localize to the same compartment when observed simultaneously.

**Figure supplement 4**. Jen1 also co-localizes to the TGN in *rod1Δ* mutant cells.

**Figure supplement 5**. Quantification of Sec7-mCh puncta that are also Jen1-GFP positive in WT and *rod1Δ* cells.

**Figure supplement 6**. Sec7 and Vps17 localize to distinct compartments in *rod1Δ* cells.

**Figure supplement 7**. Sec7 and Vps17 localize to distinct compartments in *gga1Δgga2Δ* cells.

**Figure supplement 8**. Dip5-GFP traffics through the TGN in the course of its endocytosis in WT cells.

**Figure supplement 9**. Deletions of *GGA1* and *GGA2*, encoding redundant Golgi-localized clathrin adaptor proteins, alter Dip5 trafficking to the vacuole after endocytosis.

after endocytosis in the *gga1Δ gga2Δ* mutant than in WT cells (***Figure 4H,I***), thus mimicking the defects observed in the *rod1Δ* mutant (***Figure 1*** and ***Figure 4B,C***). Again, this was not due to a defect in organelle identity caused by the deletion of the *GGA1* and *GGA2* genes because Sec7 and Vps17 appeared in distinct compartments in this strain (***Figure 4—figure supplement 7***). These results showed that TGN-to-endosome sorting is also critical for the post-endocytic targeting of Jen1 to the vacuole. Of note, we also obtained evidence that another transporter, the dicarboxylic amino acid transporter Dip5, also transiently co-localizes with the TGN during its substrate-induced endocytosis (***Figure 4—figure supplement 8***), and that its vacuolar sorting also depends on the Gga1/Gga2 proteins (***Figure 4—figure supplement 9***). Therefore, the post-endocytic control of transporter fate at the TGN is not restricted to Jen1 but is likely to be a more general mechanism.

## Dynamics of Jen1 ubiquitylation in the endocytic pathway

We showed that Jen1 ubiquitylation is required for its internalization at the plasma membrane (see ***Figure 2***). Since we identified an additional role for Rod1, which we previously showed to be critical for Jen1 ubiquitylation, in Jen1 sorting at the TGN, we hypothesized that transporter ubiquitylation may be remodeled between these two locations. This was further suggested by the observation that in retrograde mutants, in which Jen1 is internalized but not degraded, Jen1 ubiquitylation disappeared over time after glucose addition (see ***Figure 4F***). Indeed, in the *vps52Δ* mutant, in which Jen1 departs from the plasma membrane in response to glucose but cannot reach the TGN, we observed a sharp decrease in Jen1 ubiquitylation after 20 min of glucose treatment (***Figure 5A***; see also ***Figure 4F***).

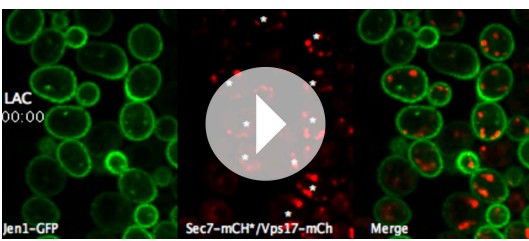

**Video 6**. Uncropped video corresponding to *Figure 4A*. Jen1-GFP co-localizes with the endosomal marker Vps17-mCh and the TGN marker Sec7-mCh during its trafficking to the vacuole in wild-type cells. WT cells expressing either both Jen1-GFP and Sec7-mCh (indicated with a white asterisk on the first image), or both Jen1-GFP and Vps17-mCh, were grown in lactate medium and visualized simultaneously for 45 min after glucose addition. A merge of Jen1-GFP fluorescence (left panel) and Vps17-mCh/Sec7-mCh fluorescence (middle panel) is shown on the right panel.

**Video 7**. Uncropped video corresponding to *Figure 4B*. Jen1-GFP co-localizes with the endosomal marker Vps17-mCh and the TGN marker Sec7-mCh during its trafficking to the vacuole in *rod1Δ* cells. *rod1Δ* cells expressing either both Jen1-GFP and Sec7-mCh (indicated with a white asterisk on the first image), or both Jen1-GFP and Vps17-mCh, were grown in lactate medium and visualized simultaneously for 45 min after glucose addition. A merge of Jen1-GFP fluorescence (left panel) and Vps17-mCh/Sec7-mCh fluorescence (middle panel) is shown on the right panel.

Because Jen1 expression is repressed by glucose (*Bojunga and Entian, 1999*), the progressive loss of Jen1 ubiquitylated species was likely due to a deubiquitylation event occurring during the progression of Jen1 from the plasma membrane to the TGN. We then performed the same experiment in a *vrp1Δ* (*end5Δ*) mutant, in which endocytosis is blocked due to the deletion of the gene encoding the yeast WASP-interacting protein (WIP) homologue, verprolin (*Munn et al., 1995*). In this mutant, however, Jen1 displayed a robust ubiquitylation pattern, indicating that the disappearance of Jen1 ubiquitylated species was not due to an intrinsic lability of ubiquitin-conjugated Jen1, and that this remodeling occurred post-internalization (*Figure 5A*).

Because we showed that Jen1 trafficking to the vacuole requires its prior targeting at the TGN, where Rod1 is localized, we hypothesized that Jen1 might be re-ubiquitylated at this compartment after endocytosis. In agreement with this hypothesis, we observed that in the *gga1Δgga2Δ* strain, in which Jen1 transiently accumulates at the TGN (see *Figure 4H,I*), Jen1 also accumulated in an ubiquitylated state, suggesting that retaining Jen1 at the TGN leads to its massive ubiquitylation (*Figure 5B*, *Figure 5—figure supplement 1*). To confirm that this was due to the ubiquitylation of Jen1 molecules originating from endosomes, we used a triple *ypt6Δ gga1Δ gga2Δ* mutant in which retrograde trafficking is abolished. In this mutant, Jen1 ubiquitylation was similar to that observed in a single *ypt6Δ* mutant, showing that the accumulation of ubiquitylated species of Jen1 at the TGN observed in the *gga1Δ gga2Δ* mutants indeed required retrograde trafficking (*Figure 5B*). In further support of an ubiquitylation event occurring at the TGN, we also observed a glucose-induced recruitment of the ubiquitin ligase Rsp5 to this compartment in response to glucose (*Figure 5C*). Using three-color imaging, we also showed a co-localization

of Rsp5 with Jen1 and Rod1 at the TGN (*Figure 5D,E*). Altogether, these data strongly suggest that Jen1 is ubiquitylated at the plasma membrane and is deubiquitylated after its internalization, and lead us to propose that a second, glucose-regulated ubiquitylation step at the TGN involving Rsp5 and Rod1 may control Jen1 progression from the TGN to the vacuole.

## The control of Jen1 trafficking at the TGN allows the recycling of internalized transporters back to the cell membrane

The yeast TGN is a well-established compartment from which cargoes can recycle to the plasma membrane (*Holthuis et al., 1998*; *Conibear et al., 2003*; *Hettema et al., 2003*; *Reggiori et al., 2003*). We thus reasoned that a control of the post-endocytic sorting of transporters at the TGN might allow transporter recycling to the cell surface upon the sudden disappearance of the signal triggering endocytosis. To test this prediction, we treated lactate-grown cells expressing Jen1-GFP with a short pulse of glucose (10 min), which was sufficient to drive Jen1-GFP internalization, and then followed Jen1-GFP trafficking upon glucose removal. Using this protocol, Jen1 redistributed back to the plasma

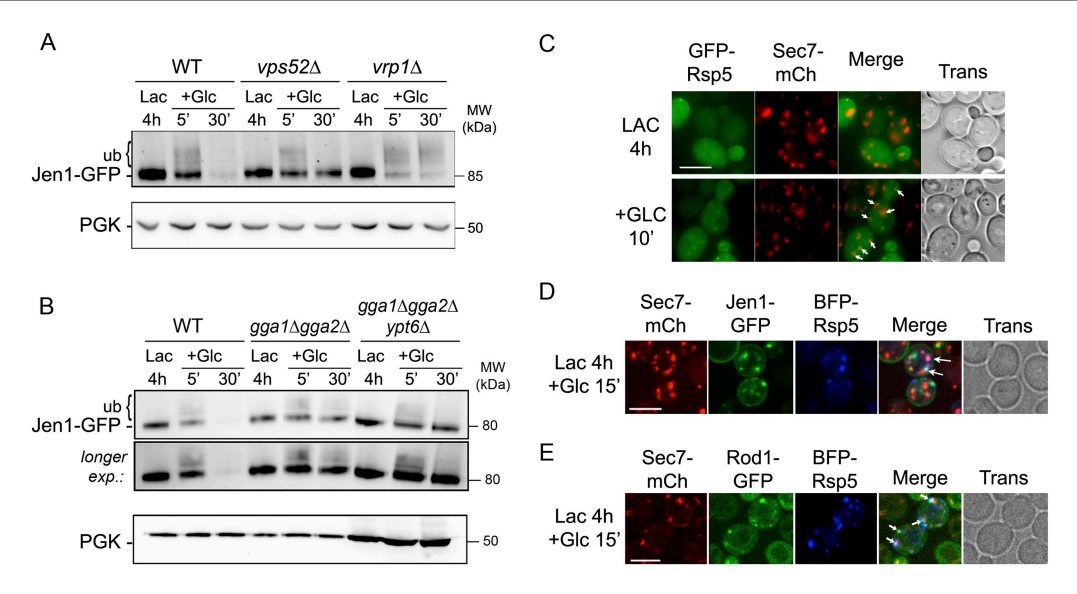

**Figure 5**. The prolonged presence of glucose is required for the full endocytosis of Jen1. (**A**) Jen1 is deubiquitylated after endocytosis. WT (ySL1150), *vps52Δ* (ySL1369) and *vrp1Δ* (ySL1610) cells expressing Jen1-GFP were grown in lactate medium and treated with glucose. Crude extracts were prepared at the indicated times and were immunoblotted with the indicated antibodies. In contrast to the situation in the *vps52Δ* mutant, in which Jen1 ubiquitylation vanishes over time, Jen1 ubiquitylation remains stable in the endocytic mutant *vrp1Δ*. (**B**) Jen1 ubiquitylation at the TGN requires retrograde sorting from endosomes to the TGN. WT (ySL1636), *gga1Δgga2Δ* (ySL1638) and *ypt6Δ gga1Δ gga2Δ* (ySL1639) cells expressing Jen1-GFP were grown in lactate medium and treated with glucose for the indicated time. Crude extracts were prepared and immunoblotted with antibodies against GFP. An increased ubiquitylation of Jen1-GFP is observed in the *gga1Δgga2Δ* mutant, that is abolished upon the additional deletion of *YPT6*. Note that Jen1-GFP is expressed at a lower level in the triple *ypt6Δ gga1Δ gga2Δ* mutant, therefore samples were loaded so that a comparable signal is observed in each strain. (**C**) A fraction of the ubiquitin ligase Rsp5 re-localizes to the TGN upon glucose addition. Cells (ySL1011) expressing both Sec7-mCh and GFP-Rsp5 were imaged after growth on lactate medium, and 10 min after glucose addition. Scale bar = 5 μm. (**D**) Rsp5 co-localizes with Jen1 at the TGN. Cells (ySL1622) expressing Sec7-mCh, Jen1-GFP and BFP-Rsp5 were grown in lactate medium, and imaged 15 min after glucose addition. Scale bar = 5 μm. (**E**) Rsp5 co-localizes with Rod1 at the TGN. Cells (ySL1625) expressing Sec7-mCh, Rod1-GFP and BFP-Rsp5 were grown in lactate medium, and imaged 15 min after glucose addition. Scale bar = 5 μm.

The following figure supplement is available for figure 5:

**Figure supplement 1**. Jen1 accumulates in an ubiquitylated form in the *gga1Δ gga2Δ* mutant.

membrane within 20 min (**Figure 6A**), showing that Jen1 endocytosis was reversible upon glucose removal. This change in localization coincided with a loss of Jen1 ubiquitylation, suggesting that a continuous presence of glucose is required for the persistent ubiquitylation of Jen1 and its degradation (**Figure 6B**). The same experiment performed in the *vrp1Δ* mutant, in which endocytosis is blocked, also led to a loss of Jen1 ubiquitylation, showing it is not due to Jen1 degradation but rather to a rapid deubiquitylation event following glucose removal, regardless of Jen1 localization (**Figure 6B**). Of note, Jen1 no longer recycled to the plasma membrane in the *vps52Δ* strain in these conditions (**Figure 6C** and **Video 8**), indicating that Jen1 recycling requires a functional endosome-to-TGN retrograde pathway. This also confirmed that the signal recovered at the plasma membrane upon glucose removal in WT cells (see **Figure 6A**) did not originate from Jen1 neosynthesis. Therefore, a second, glucose-based sorting event occurs after internalization and takes place at the TGN. Accordingly, Jen1 recycling led to its polarized redistribution to the cell buds (**Figure 6C** and **Video 8**), similarly to the exocytic SNARE Snc1 and the chitin synthase Chs3, which localize to the cell membrane in a polarized fashion after endocytosis and recycling through the TGN (**Holthuis et al., 1998**; **Lewis et al., 2000**). Interestingly, Jen1 recycling to the cell membrane upon glucose removal also

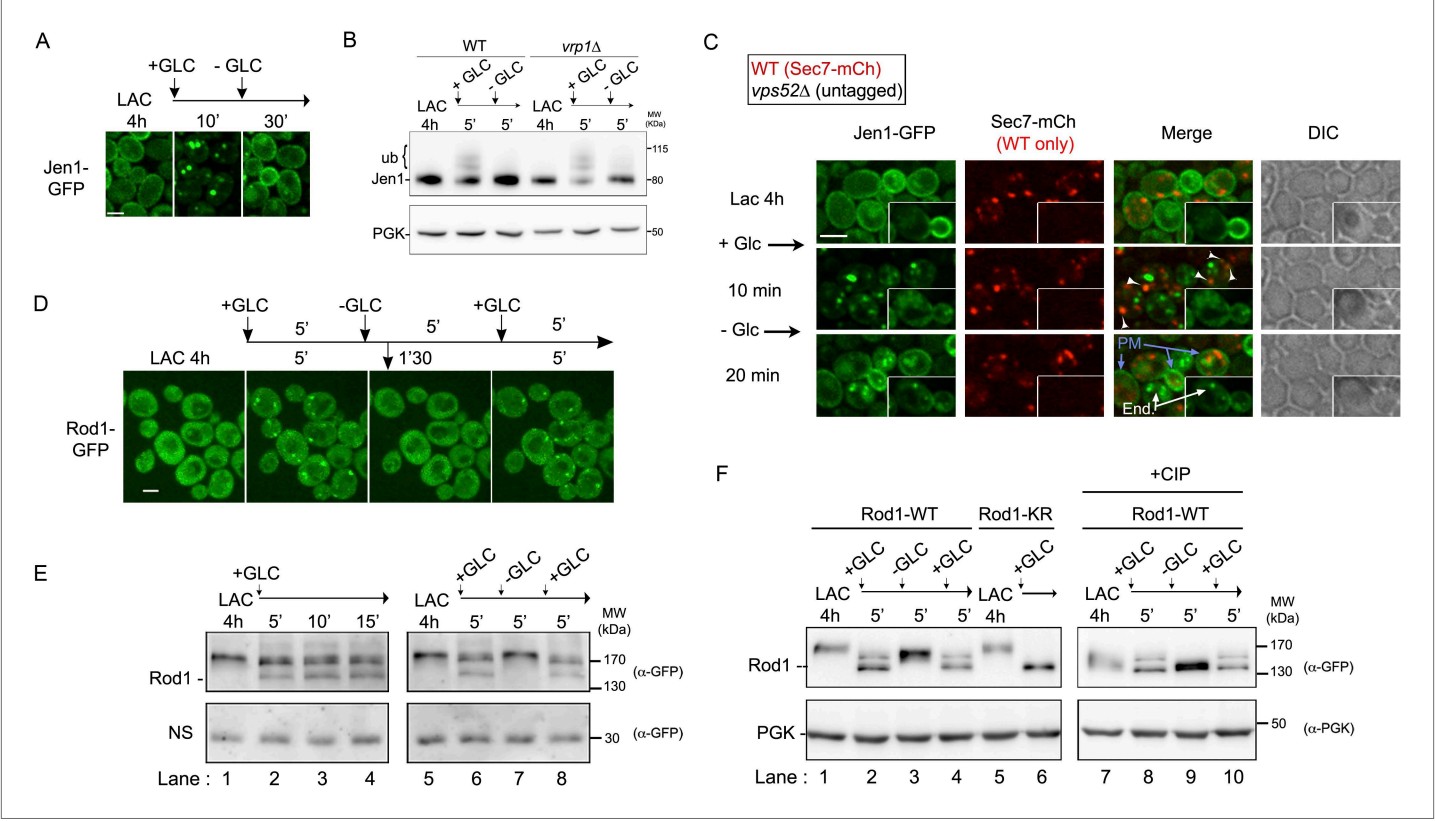

**Figure 6**. The control of Jen1 trafficking at the TGN allows the recycling of internalized transporters back to the cell membrane. (**A**) Jen1 endocytosis is reversible upon glucose removal. Lactate-grown WT cells expressing Jen1-GFP (ySL1150) were injected into a microfluidics device in lactate medium. Cells were then imaged over time, before and after 10 min glucose addition, and then 20 min after glucose removal. Scale bar = 2.5 µm. (**B**) The persistence of the glucose signal is required to maintain Jen1 ubiquitylation. Lactate-grown WT cells expressing Jen1-GFP (ySL1150) were treated with glucose for 5 min. Cells were then briefly centrifuged and incubated back into lactate medium for 5 min. Crude extracts were prepared at each step and were immunoblotted with the indicated antibodies. The high-molecular weight species of Jen1 appearing upon glucose addition correspond to ubiquitylated Jen1 (see **Becuwe et al., 2012b**) (see also **Figure 2B**). The same extracts prepared from *vrp1Δ* cells show that Jen1 deubiquitylation observed in WT cells is not due to Jen1 degradation but to an active deubiquitylation process. (**C**) Lactate-grown WT (ySL1165) cells expressing both Jen1-GFP and Sec7-mCh, and *vps52Δ* (ySL1369) cells expressing only Jen1-GFP were co-injected into the microfluidics device in lactate medium, before glucose was added for 10 min and then removed. Cells were imaged at the indicated times. Arrowheads indicate examples of co-localization of Jen1-GFP and Sec7-mCh in WT cells, and arrows show the plasma membrane localization of Jen1-GFP after glucose removal in WT cells. Note the polarized distribution of Jen1-GFP after recycling in WT cells, but not in *vps52Δ* cells (inset). *PM*, plasma membrane; end: endosomes. Scale bar = 2.5 µm. See also **Video 8**. (**D**) Jen1 recycling correlates with the loss of Rod1 localization to the TGN. Lactate-grown cells (ySL542) expressing Rod1-GFP were injected into a microfluidics device in lactate medium. Cells were then subjected to 5-min pulses of glucose addition/removal and imaged simultaneously. Only the first three cycles are shown here. Scale bar = 2.5 µm. See also **Video 9**. (**E–F**) Dynamics of Rod1 post-translational modifications upon glucose/lactate cycles. Lactate-grown WT cells expressing a plasmid-borne Rod1-GFP were treated with glucose for 15', or subjected to glucose addition/removal as indicated. Crude extracts were prepared at each step and were immunoblotted with the indicated antibodies. A non-specific (NS) cross-reacting band is used as a loading control. The high-molecular weight species of Rod1-GFP observed in the lactate samples correspond to phosphorylation (panel **E**, lanes 1, 5 and 7), because phosphatase treatment (CIP) abolishes the migration shift (panel **F**, lanes 7 and 9). The glucose-induced doublet (panel **E**, lanes 2, 3, 4, 6 and 8) corresponds to ubiquitylated (higher band) and non-ubiquitylated (lower band) forms of Rod1, because mutation of its ubiquitylation sites (using a plasmid encoding Rod1-KR) abolishes the migration shift (panel **F**, lane 6), as described previously (**Becuwe et al., 2012b**). PGK is used as a loading control in panel **F**.

correlated with the loss of Rod1 localization at the TGN (**Figure 6D**). Rod1 localization was indeed extremely responsive to glucose treatments, and applying repeated glucose/lactate cycles revealed a robust sensor-like response of Rod1 regarding its localization (**Figure 6D** and **Video 9**) and post-translational modifications (**Figure 6E,F**). Therefore, Jen1 ubiquitylation and endocytosis can be re-evaluated at the TGN after internalization upon disappearance of the endocytic stimulus, and this correlates with a rapid glucose-induced remodeling of Rod1 post-translational modifications and its redistribution from the TGN to the cytosol.

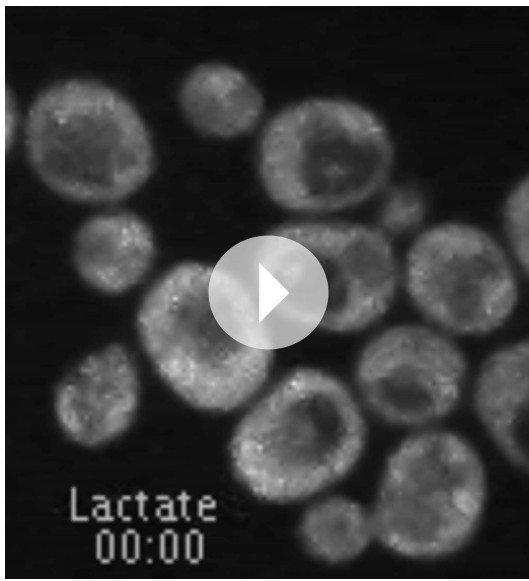

**Video 8**. Jen1 recycles back to the plasma membrane via the TGN. WT cells expressing both Jen1-GFP and Sec7-mCh, and *vps52Δ* cells expressing only Jen1-GFP were grown 4 hr in lactate medium and observed simultaneously for 10 min of glucose addition and 20 min after glucose removal. Co-localization between Jen1-GFP and Sec7-mCh in WT cells is indicated by white arrows in the merge panel (second on the right). See also *Figure 6C*.

**Video 9**. Jen1 recycling correlates with the loss of Rod1-localization to the TGN. WT cells expressing Rod1-GFP were grown for 4 hr in lactate medium and observed during 3 cycles of glucose addition/removal (5-min pulses). See also *Figure 6D*.

## Rod1 also controls the exit of neosynthesized Jen1 from the TGN to the vacuole

Neosynthesized transporters traffic through the TGN en route to the plasma membrane in the secretory pathway. However, many transporters can escape this pathway and traffic from the TGN to the vacuole when they are expressed in conditions that would normally trigger their endocytosis at the plasma membrane (for review, see *Haguenauer-Tsapis and André, 2004*). We used a galactose-inducible Jen1-GFP construct, which allows Jen1-GFP targeting to the cell surface in galactose medium (*Figure 7A*) (*Paiva et al., 2009*), and followed the effect of glucose during Jen1-GFP targeting to the membrane (see *Figure 7B*, top). We observed that the addition of glucose perturbed Jen1 sorting in the secretory pathway, as Jen1 was targeted to the vacuole instead of the plasma membrane in WT cells (*Figure 7B*, *Video 10*). This localization did not involve Jen1 targeting to the plasma membrane and its endocytosis, because Jen1 behaved similarly in the endocytic mutant *vrp1Δ* (*Figure 7—figure supplement 1*). In the *rod1Δ* mutant, however, Jen1 was no longer targeted to the vacuole, but instead reached the cell surface (*Figure 7B*, *Video 10*), showing that Rod1 is required to divert Jen1 trafficking from the secretory pathway to the vacuole. Noteworthy, Rod1 was again recruited to the TGN in these conditions upon glucose addition (*Figure 7C*).

Because Rod1 contributes to Jen1 ubiquitylation in response to glucose (*Becuwe et al., 2012b*), we tested whether the inability of Jen1 to be sorted from the TGN to the vacuole in the *rod1Δ* mutant was due to a lack of Jen1 ubiquitylation. In agreement with this hypothesis, a non-ubiquitylatable Jen1-KR-GFP construct (see

*Figure 2*) was targeted to the plasma membrane despite the presence of glucose (*Figure 7D*), showing that Jen1 ubiquitylation is critical for its rerouting from the secretory pathway to the vacuolar pathway. A similar result was obtained in a hypomorphic mutant of *RSP5*, *npi1*, showing that the vacuolar sorting of neosynthesized Jen1 from the TGN to the vacuole also involved Rsp5 (*Figure 7E*). Furthermore, deletion of the genes encoding the ubiquitin-binding, clathrin adaptor proteins Gga1 and Gga2 proteins also led to a defect in the vacuolar targeting of neosynthesized Jen1 in the presence of glucose (*Figure 7F*). Altogether, these data indicate that the observed glucose-induced recruitment of Rod1 and Rsp5 to the TGN promotes Jen1 ubiquitylation and its exit from the TGN to the vacuole in a Gga1-Gga2 dependent manner.

## Discussion

Arrestin-related proteins have emerged as key players in the regulation of transporter endocytosis and degradation by nutrient signaling pathways in yeast and mammalian cells (*O'Donnell, 2012*; *Wu et al., 2013*). In the present study, the use of microfluidics-assisted live cell imaging allowed us to follow, for the first time, the dynamics of cargo trafficking in response to variations in the presence of the endocytic stimulus, as well as the localization of the ART protein Rod1 in these conditions. This revealed a

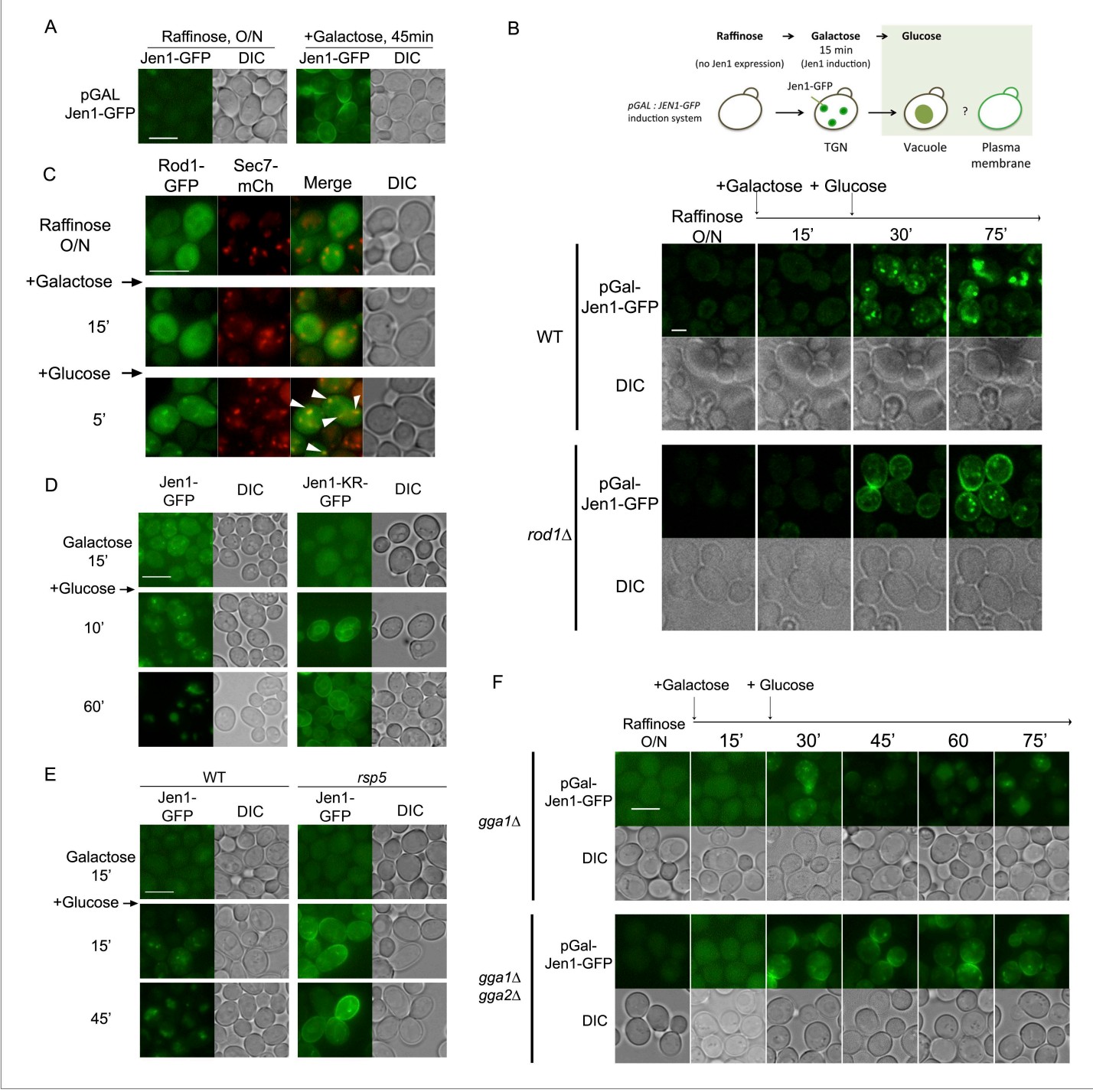

**Figure 7**. Rod1 promotes Jen1 exit from the secretory pathway to the vacuole. (**A**) A galactose-inducible Jen1-GFP is targeted to the plasma membrane in galactose medium. WT cells (ySL1083) expressing a galactose-inducible Jen1-GFP were grown in raffinose medium overnight, and transferred to galactose medium for 45 min to allow Jen1-GFP expression and targeting to the plasma membrane. Scale bar = 5 μm. (**B**) Rod1 is required for the glucose-induced retargeting of Jen1 from the secretory pathway to the vacuole. Top, outline of the experiment. A 15-min pulse of galactose allows the synthesis of Jen1-GFP, and the effect of glucose on the sorting of neosynthesized Jen1-GFP is then monitored. Bottom, WT (ySL1083) and *rod1Δ* (ySL781) cells both expressing a galactose-inducible Jen1-GFP were grown overnight on raffinose medium. After 15 min of galactose induction, glucose was then added to the cells and Jen1-GFP fluorescence was followed over time. Scale bar = 2.5 μm. See also *Video 10*. Scale bar = 2.5 μm. Note that the sorting of neosynthesized Jen1 to the vacuole in response to glucose does not require targeting to the plasma membrane and endocytosis, see *Figure 7—figure supplement 1*. (**C**) Rod1 localizes to the TGN when transferred from galactose to glucose medium. Cells (ySL638) co-expressing

*Figure 7. Continued on next page*

*Figure 7. Continued*

Rod1-GFP and Sec7-mCh were grown overnight on raffinose medium, and observed 15 min after galactose addition and then and 5 min after glucose addition. Arrowheads indicate co-localization events between Rod1-GFP and Sec7-mCH. Scale bar = 5 µm. (**D**) The non-ubiquitylatable Jen1-KR-GFP construct fails to be targeted from the secretory pathway to the vacuole in response to glucose. WT cells expressing either a plasmid-encoded, galactose inducible Jen1-GFP (ySL1083) or the mutant Jen1-KR-GFP (ySL1339) construct were grown as in *Figure 7B*, and imaged at the indicated times. Note that the Jen1-KR-GFP presents defects in ER exit upon synthesis (ER labeling observed throughout the experiment) but still manages to reach the plasma membrane, and is not associated with the vacuole. Scale bar = 5 µm. (**E**) The sorting of neosynthesized Jen1 to the vacuole in response to glucose requires Rsp5. WT (ySL1083) and *npi1* (a hypomorphic *rsp5* mutant; ySL1556) cells both expressing a galactose-inducible Jen1-GFP were grown overnight on raffinose medium. After 15 min of galactose induction, glucose was then added to the cells and Jen1-GFP fluorescence was followed over time. Scale bar = 5 µm. (**F**) The Golgi-localized clathrin adaptor proteins, Gga1 and Gga2, are required for the glucose-induced retargeting of Jen1 from the secretory pathway to the vacuole. Strains *gga1Δgga2Δ* (ySL1311) or *gga1Δgga2Δ* expressing Gga2-HA (ySL1310), used as a positive control, both expressing Jen1-GFP from a plasmid-encoded, galactose-inducible construct were induced as in *Figure 7B*, and imaged at the indicated times. Scale bar = 5 µm.

The following figure supplement is available for figure 7:

**Figure supplement 1**. The sorting of neosynthesized Jen1 to the vacuole in response to glucose does not require targeting to the plasma membrane and endocytosis.

dual function of the ART protein Rod1 in the control of transporter endocytosis and degradation by glucose availability (*Figure 8*, left and middle panel). First, Rod1 exerts a function at the plasma membrane, as it is essential for the endocytosis of the glycerol/proton symporter Stl1, and required for the efficient internalization of Jen1 upon glucose treatment (*Figure 1*). Second, Rod1 also controls transporter sorting after endocytosis, a step which takes place at the trans-Golgi network (*Figure 8*, middle panel). Indeed, Jen1 localizes to the TGN after endocytosis, and both (i) the endosome-to-Golgi retrograde pathway and (ii) the Gga1/2-dependent, Golgi-to-vacuole pathway are required for the vacuolar delivery of Jen1, indicating a critical role for the TGN in transporter degradation (*Figure 4*). Moreover, we showed that the endocytosis of the unrelated amino acid transporter, Dip5, in response to its substrate also involves a step at the TGN as well as the GGA proteins. These results shed a new light on previous studies showing the involvement of GGA proteins in the endocytic trafficking of Gap1, the general amino-acid permease (*Scott et al., 2004*; *Lauwers et al., 2009*), which could be explained by a similar control of Gap1 fate at the TGN. The yeast TGN therefore appears as a critical organelle for the post-endocytic sorting of transporters. This new level of regulation provides an opportunity to re-evaluate, after internalization, the commitment of transporters to the degradation pathway. Indeed, upon glucose removal, we show that internalized Jen1 recycles to the cell surface after it has reached the TGN (*Figure 6*; *Figure 8*, right panel).

Whereas the involvement of the TGN in cargo recycling to the cell membrane has already been documented (*Holthuis et al., 1998*; *Conibear et al., 2003*; *Hettema et al., 2003*; *Reggiori et al., 2003*), little information was available concerning the regulation of cargo recycling in response to external signals (*Strochlic et al., 2008*). Here, we propose that external signals can remodel transporter ubiquitylation at the TGN through the regulation of ART proteins, and therefore control transporter targeting towards the recycling or degradation pathways. Indeed, Jen1 ubiquitylation is highly dynamic and can be re-evaluated after internalization. Through the use of trafficking mutants, we observed a progressive loss of Jen1 ubiquitylation between the plasma membrane and the TGN, even in the presence of glucose (*Figure 5A*) suggesting that Jen1 ubiquitylation status is reset between these compartments, likely through the action of deubiquitylating enzymes that are yet to be identified. This may set the stage for a second Jen1 ubiquitylation event at the TGN, that would allow cells to control transporter progression to the vacuole in a glucose- and Rod1-dependent manner (*Figure 8*, right panel). This is further suggested by the fact that Jen recycling upon glucose removal coincides with the loss of its ubiquitylation (*Figure 6B*). Therefore, the continuous presence of glucose is not only required to trigger Jen1 internalization, but also to maintain Jen1 in the endocytic pathway and prevent its recycling. Altogether, our data shed light on the integrated nature of the endocytic decision, and the multiple places where signaling can impact on transporter sorting through the ART/Rsp5 network.

This regulation recalls that of the β2-adrenergic receptor, for which the multifunctional adaptor protein, β-arrestin2, promotes receptor ubiquitylation after its endocytosis through the recruitment of the ubiquitin ligase Nedd4 (*Shenoy et al., 2008*) but also regulates receptor deubiquitylation by the

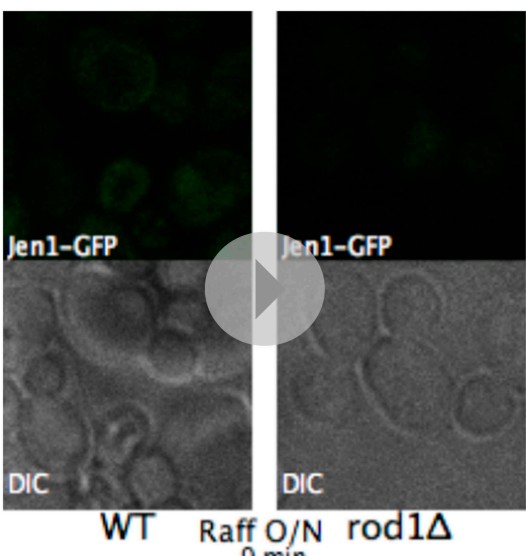

**Video 10**. Rod1 is required for the glucose-induced retargeting of Jen1 from the secretory pathway to the vacuole. WT cells (left panel) and *rod1Δ* cells (right panel) expressing Jen1-GFP under a galactose-inducible promoter were grown in raffinose medium overnight and simultaneously observed for 15 min during galactose induction and then 45 min after glucose addition. See also *Figure 7B*.

deubiquitylating enzymes USP33 and USP20, hence its recycling, depending on the duration of the stimulation (*Berthouze et al., 2009*) (reviewed in *Kommaddi and Shenoy, 2013*). However, in this specific case, the way by which agonist exposure is perceived and conveyed to edit the ubiquitin signal on the cargo is not fully understood. Here, we propose that a glucose-induced recruitment of the arrestin-related protein Rod1 to the TGN curtails Jen1 recycling and promotes its vacuolar sorting, leading to its degradation. This re-localization relies on rapid glucose-induced changes in Rod1 post-translational modifications through a mechanism that we previously established (*Becuwe et al., 2012b*), and both are reversible upon glucose removal (*Figure 6D–F*). The latter situation coincides with a loss of Jen1 ubiquitylation and its recycling, which may be caused by its inability to be re-ubiquitylated at the TGN because Rod1 is no longer at this compartment. The mechanism of the dynamic recruitment of Rod1 to the TGN will need to be further explored. Rod1 is so far the only arrestin-related protein shown to display a signal-induced localization to the TGN, although other arrestin-related proteins also localize to internal compartments at steady-state, such as the TGN or endosomes, both in yeast (*Lin et al., 2008*; *O'Donnell et al., 2010*) and mammalian cells (*Vina-Vilaseca et al., 2011*; *Han et al., 2013*). We also showed a dynamic recruitment of Rsp5 to the TGN in response to glucose, which is line with our model of a re-ubiquitylation of Jen1 at the TGN after endocytosis. The previously reported interaction between Rsp5 and Sec7 (*Dehring et al., 2008*) may provide insights into the molecular basis of Rsp5 recruitment to this organelle.

Noteworthy, the use of the same machinery to regulate transporter trafficking at the plasma membrane and the TGN also allows an integrated control of transporter sorting both during secretion and after endocytosis at the same location (*Figure 7*; *Figure 8*, middle panel). This may be critical to prevent neosynthesized cargoes transiting in the secretory pathway from reaching the cell surface in conditions that induce endocytosis. Intriguingly, the ART-related proteins Bul1 and Bul2 were shown to control the sorting of neosynthesized Gap1 between the Golgi and the vacuole (*Helliwell et al., 2001*; *Soetens et al., 2001*; *Risinger and Kaiser, 2008*), and then later found to be involved in Gap1 down-regulation, suggestive of a role at the plasma membrane (*Risinger and Kaiser, 2008*; *Merhi and André, 2012*). In the light of our data, it appears likely that the Bul1/Bul2 proteins may in fact act at the TGN to control both of these sorting events. The study of the subcellular localization of the Bul1/Bul2 proteins may provide important information in this regard. Interestingly, recent studies in the evolutionary distant yeast *Schizosaccharomyces pombe* also point to a role of the arrestin-related protein Any1 (also named Arn1) in transporter trafficking at multiple places in the cell (*Nakase et al., 2013*; *Nakashima et al., 2014*). Finally, data on the ART protein Aly2 suggest both a role in endocytosis and endosomal recycling depending on the transporter or the physiological condition considered (*Hatakeyama et al., 2010*; *O'Donnell et al., 2010*; *Crapeau et al., 2014*).

In metazoans, recycling endosomes are specific organelles dedicated to the recycling of internalized proteins to the plasma membrane (*Sheff et al., 1999*). They notably contribute to epithelial cell polarity (*Golachowska et al., 2010*) and membrane remodeling in neurons (*Schmidt and Haucke, 2007*) but the regulation of cargo recycling is poorly understood at the molecular level (*Maxfield and McGraw, 2004*). So far, a role of the TGN in cargo recycling has been only marginally studied (*Csaba et al., 2007*; *Escola et al., 2010*; *Cheng and Filardo, 2012*). The finding that Ypt32, a yeast TGN-localized Rab protein involved in recycling from the TGN to the cell surface, localizes to recycling

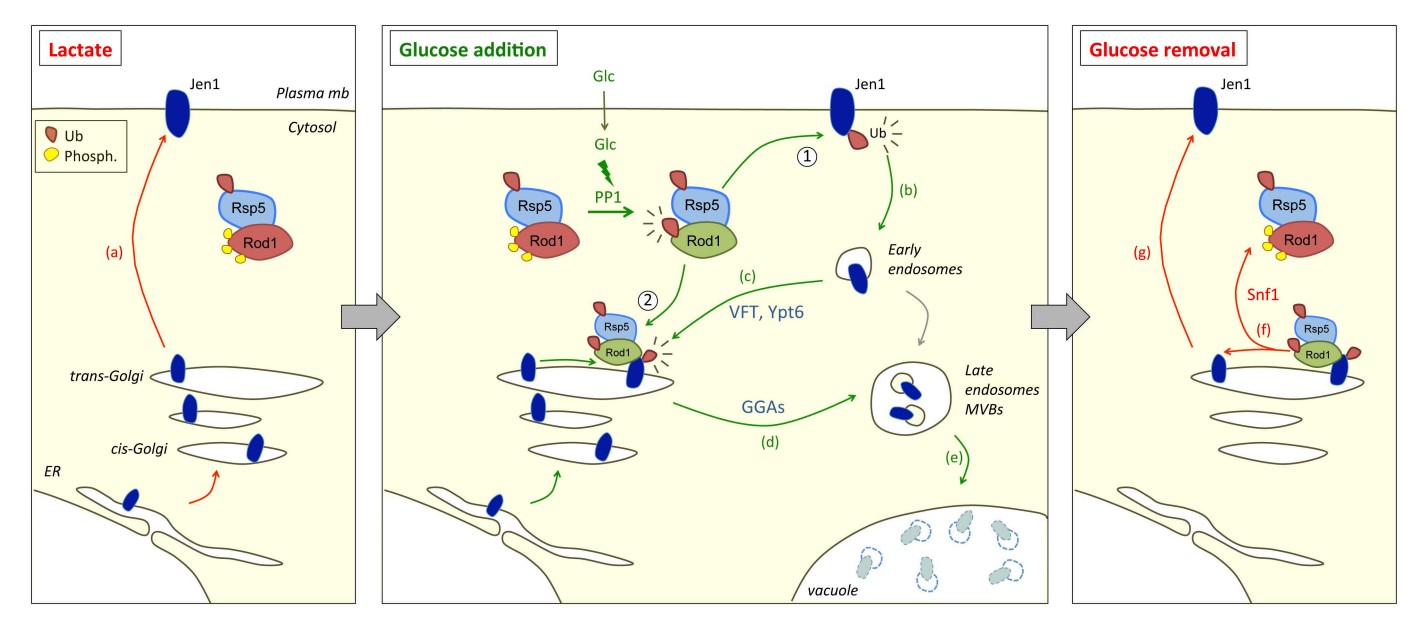

**Figure 8**. Working model for the dual function of Rod1 in the regulation of transporter endocytosis and recycling. Left, in lactate medium, Jen1 is synthesized and targeted to the plasma membrane. Although Rod1 interacts with Rsp5 (**Becuwe et al., 2012b**), it is inactive (phosphorylated) and cytosolic. Middle, In the presence of glucose, Rod1 is activated by a protein phosphatase 1 (PP1)-dependent dephosphorylation and its subsequent Rsp5-mediated ubiquitylation (**Becuwe et al., 2012b**). Rod1 promotes transporter internalization at the plasma membrane (1), but also localizes to the TGN (2). There, Rod1 controls the fate of internalized transporters that have been retrograde-trafficked by a VFT- and Ypt6-dependent pathway, and also that of transporters coming from the secretory pathway. Transporter sorting from the TGN to the vacuole requires Rod1 and the clathrin adaptors Gga1 or Gga2 (GGAs). Right, Upon glucose removal, shortly after endocytosis is initiated, Rod1 is rephosphorylated (likely by the kinase Snf1) (**Becuwe et al., 2012b**), leading to Rod1 dissociation from the TGN and Jen1 recycling to the cell membrane.

endosomes in mammalian cells and affects transferrin receptor recycling (**Kail et al., 2005**) suggests that recycling from endosomes and the TGN may be evolutionary related. Our work in yeast further indicates that this may be an ancestral mode of regulation, and provides insights into the molecular mechanism by which it is achieved. This study also illustrates an underappreciated role of the TGN in the integrated regulation of recycling and secretion that should be considered for future studies.

## Materials and methods

### Yeast strains, transformation and growth conditions

Strains are listed and detailed in **Supplementary file 1**. All strains are derivatives of the BY4741/2 strains, except for the *gga1Δ GGA2-HA* and *gga1Δ gga2Δ* strains, which were kindly provided by Prof. R Piper, University of Iowa, Iowa City, USA (**Scott et al., 2004**) and the *gga1Δgga2Δ* and *gga1Δgga2Δ ypt6Δ* strains (23344c background) that were kindly provided by Prof. B André, Université Libre de Bruxelles, Belgium (**Lauwers et al., 2009**). The *9-arrestin* mutant was kindly provided by Dr H Pelham (MRC Laboratory of Molecular Biology, Cambridge, UK) (**Nikko and Pelham, 2009**). Yeast was transformed by standard lithium acetate/polyethylene glycol procedure. Cells were grown in yeast extract/peptone/glucose (YPD) rich medium, or in synthetic complete (SC) medium containing 2% (wt/vol) Glc, or 0.5% (vol/vol) Na-lactate (pH 5.0) (Sigma-Aldrich, Lyon, France). For lactate inductions, cells were grown overnight in SC-Glc, harvested in early exponential phase ($A_{600}$ = 0.3), resuspended in the same volume of SC-lactate and grown for 4 hr ($A_{600}$ = 0.5), before the addition of glucose (2% wt/vol, final concentration). For the observation of Stl1-GFP, cells were grown for 2 hr in lactate medium, and glycerol was then added (3% vol/vol) for 2 hr to induce Stl1-GFP expression and targeting to the plasma membrane. For galactose induction, cells were precultured in SC-Glc medium, and grown overnight to early exponential phase ($A_{600}$ = 0.3) in SC medium containing 2% raffinose (wt/vol) and

0.02% Glc (wt/vol) to initiate growth. Galactose was then added at a final concentration of 2% (wt/vol) and cells were grown for the indicated times. Chase/endocytosis was started by adding glucose to a final concentration of 2% (wt/vol) and incubating for the indicated times. Latrunculin A (Sigma) was used a final concentration of 0.2 mM.

## Plasmids and constructs

For the Jen1-KR-GFP mutagenesis, the *JEN1* ORF and its promoter were first amplified by PCR from BY4741 genomic DNA (using oligonucleotides oSL337/oSL338), the fragment was digested with SacI/SpeI, cloned at SacI/SpeI sites into a pRS416-based vector containing GFP (pRHT140, lab collection), and sequenced (pSL161). A synthetic gene encoding the Jen1-KR mutant was generated (Eurofins MWG Operon, Ebersberg, Germany), amplified by PCR (oSL371/oSL394), cloned by gap-repair in yeast into pSL161 digested with HindIII/SpeI, and sequenced (pSL163). The galactose-inducible version was cloned similarly: the synthetic gene was amplified by PCR (oSL476/oSL477), and cloned by gap-repair in yeast into pRHT373 (*Becuwe et al., 2012b*) (pSL184). The plasmid encoding Rod1-GFP (*Figures 2D,4G*) was described previously (pSL93) (*Becuwe et al., 2012b*). The plasmid encoding Rod1-KR-GFP (*Figure 6F*) was generated by cloning a SacI/XmaI fragment from pSL143 (*Becuwe et al., 2012b*) into pSL93. The plasmid encoding mTag-BFP2-Rsp5 (pSL303, *Figure 5D,E*) was constructed by PCR amplification of the mTagBFP2 sequence (TagBFP2-N, Evrogen JSC, Moscow, Russia) (oSL652/oSL653), digestion and cloning at XbaI/NotI sites in place of GFP in pSL19 (p415-pADH-GFP-Rsp5, *Leon et al., 2008*).

## Total protein extracts and phosphatase treatment

For total protein extracts, trichloroacetic acid (TCA; Sigma-Aldrich) was added directly in the culture to a final concentration of 10% (vol/vol), and cells were precipitated on ice for 10 min. Cells were then harvested by centrifugation for 1 min at room temperature at 16,000×*g*, then lysed with glass beads in a 100 µl of TCA (10%, vol/vol) for 10 min at 4°C. Beads were removed, the lysate was centrifuged for 1 min at room temperature at 16,000×*g*, and the resulting pellet was resuspended in TCA-sample buffer (Tris–HCl 50 mM pH 6.8, dithiothreitol 100 mM, SDS 2%, bromphenol blue 0.1%, glycerol 10%, containing 200 mM of unbuffered Tris solution) at a concentration of 50 µl/initial OD unit, before being denatured at 37°C for 10 min. Phosphatase treatment was performed as previously described (*Becuwe et al., 2012b*).

## Antibodies and immunoblotting

We used monoclonal antibodies raised against GFP (clones 7.1 and 13.1; Roche Diagnostics, Meylan, France), HA (clone F7; Santa Cruz Biotechnology, Dallas, TX), anti-ubiquitin antibody coupled to horseradish peroxidase (clone P4D1; Santa Cruz Biotechnology), and polyclonal antibodies against 3-phosphoglycerate kinase (PGK) (clone 22CS; Life Technologies, Saint Aubin, France). Immunoblots were acquired with the LAS-4000 imaging system (Fuji, Tokyo, Japan). Quantification was performed using ImageJ (NIH) on non-saturated blots.

## Fluorescence microscopy

Cells were mounted in synthetic complete medium with the appropriate carbon source and observed with a motorized Olympus BX-61 fluorescence microscope equipped with an Olympus PlanApo 100× oil-immersion objective (1.40 NA), a Spot 4.05 charge-coupled device camera and the MetaVue acquisition software (Molecular Devices; Sunnyvale, CA). Cells were mounted in SD medium and imaged at room temperature. GFP-tagged proteins were visualized using a Chroma GFP II filter (excitation wavelength 440–470 nm). mCh-tagged proteins were visualized using an HcRed I filter (excitation wavelength 525–575 nm). Images were processed in ImageJ (NIH) and Photoshop (Adobe, San Jose, CA) for levels.

Vacuolar staining was obtained by incubating cells with 100 µM CMAC (Life Technologies) for 10 min under agitation at 30°C, then cells were then washed twice with water before observations with a confocal microscope (see references below) equipped with a DAPI filter (450QM60).

For the microfluidics experiments, cells growing in exponential phase (DO = 0.3–0.6) were injected in a CellASIC microfluidics chamber (ref. YO4C, Merck-Millipore, Darmstadt, Germany), using the Microfluidic Perfusion Platform (ONIX), driven with the interface software ONIX-FG-SW (Merck-Millipore). Cells were trapped and maintained in a uniform plane. Normal growth conditions were reproduced by adjusting the ambient temperature at 30°C with a thermostated chamber, and by

flowing cells with the indicated culture medium at 3 psi. The microfluidics device was coupled to a DMI6000 (Leica, Buffalo Grove, IL) microscope, equipped with an oil immersion plan apochromat 100× objective NA 1,4, a QuantEM cooled EMCCD camera (Photometrics, Tucson, AZ), and a spinning-disk confocal system CSU22 (Yokogawa, Tokyo, Japan). Image resolution was 1 pixel = 149 nm. GFP-tagged proteins, mCh-tagged proteins and CMAC staining were visualized with a GFP Filter 535AF45, RFP Filter 590DF35, and DAPI Filter 450QM60 respectively. Images were acquired with MetaMorph 7 software (Molecular Devices, Sunnyvale, CA), and denoised (*Figure 1A,C,D–F*; *Figure 3*; *Figure 3A,B,F*; *Figure 6A,C,D*; *Figure 7B*; and all videos), with the Image J plugin Safir Filter (*Kervrann and Boulanger, 2006*).

For *Figure 4—figure supplement 3*, images were acquired with a Revolution xD TuCam system (Andor) equipped with a confocal scanner unit CSU-X1 (Yokogawa) and a Ti microscope with a 100x/1.4 NA objective (Nikon, Tokyo, Japan) and piloted by MetaMorph software (Molecular Devices). Simultaneous acquisitions of two channels were done using the TuCam device (Andor, Belfast, UK) equipped with 580 dicroic filter and 525/50 (green), 616/73 (red) emission filters. Detectors on the TuCam are iXon3 EMCCD camera (Andor) with 8 µm well size. Image stacks are acquired with a Pifoc (Physik Intrumente, Karlsruhe, Germany) with a z-step of 0.2 µm.

## Quantification of co-localizations and trafficking delay

Quantifications to evaluate the delayed internalization of Jen1-GFP in *rod1Δ* vs WT cells were performed manually (*Figure 1G*). The number of Jen1-containing vesicles was quantified over time in each strain by three manual counting on 30 cells, taken from three independent experiments. The total number of Jen1-GFP-labeled structures were counted for the 30 cells and divided by 30. STDEV was also calculated for each time point. Manual quantifications were also performed for co-localization event between Jen1-GFP with markers of internal compartments (Sec7-mCh and Vps17-mCh) (*Figure 4C,I*). Counting was performed three times on 20 cells, taken from three independent experiments. Over time, the percentage of Jen1-GFP-containing vesicles co-localizing with the mCherry marker (Sec7 or Vps17) was calculated for each cell and divided by the mean number of vesicles per cell. STDEV was also calculated for each time point.

## Acknowledgements

We thank Bruno André, Hugh Pelham and Robert Piper for strains; Pascal Hersen and Agnès Miermont for their initial help with microfluidics; the imaging facility of the Institut Jacques Monod (ImagoSeine) for assistance; Angela Taddei (Institut Curie, Paris) for the use of the Revolution xD TuCam microscope; and Catherine L Jackson for the use of the microfluidics device. We are indebted to Bruno André, Alenka Čopič, Thierry Galli, Pascal Hersen, Catherine L Jackson, as well as Rosine Haguenauer-Tsapis and other members of the lab for helpful discussions and comments of the manuscript.

## Additional information

### Funding

| Funder | Grant reference number | Author |
| --- | --- | --- |
| Fondation ARC pour la Recherche sur le Cancer | SFI20121205762 | Sébastien Léon |
| Ligue Contre le Cancer | Comite de Paris - RS13/75-45 | Sébastien Léon |
| Ligue Contre le Cancer | Comite de Paris - RS14/75-120 | Sébastien Léon |
| Fondation ARC pour la Recherche sur le Cancer | DOC20130606445 | Michel Becuwe |

The funders had no role in study design, data collection and interpretation, or the decision to submit the work for publication.

### Author contributions

MB, SL, Conception and design, Acquisition of data, Analysis and interpretation of data, Drafting or revising the article, Contributed unpublished essential data or reagents

## Additional files

**Supplementary file**
• Supplementary file 1. A table listing yeast strains used in this study is provided in *Supplementary file 1*.

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
