## [Decision Letter]

Thank you for sending your work entitled “Integrated control of transporter endocytosis and recycling by the Rod1-Rsp5 ubiquitylation complex” for consideration at *eLife.* Your article has been favorably evaluated by Randy Schekman (Senior editor) and 3 reviewers, one of whom, Ivan Dikic, is a member of our Board of Reviewing Editors.

The Reviewing editor and the other reviewers discussed their comments before we reached this decision, and the Reviewing editor has assembled the following comments to help you prepare a revised submission.

The manuscript describes that Rod1 acts at the TGN to route some endocytosed cell surface proteins to the vacuole via a GGA mediated pathway. This is a novel finding and distinct from roles for arrestins described in endocytosis and in traffic from the endosome to the TGN.

However, the authors need to address the following points before the manuscript can be considered for publication in *eLife*:

1) The authors speculate that Jen1 gets actively de-ubiquitylated after internalization and therefore requires another round of ubiquitylation in the TGN in order to be directed to the vacuole. As this two-step model is a central point of this manuscript they should test what happens if the ubiquitination cycle is blocked and whether in that case a localization of Jen1 to the TGN and Rod1 is still required. Or does aberrant ubiquitination bypass the TGN and/or the requirement for Rod1/Rsp5 on the way to the vacuole? The authors should look at the ubiquitination state in the GGA mutant. Would Jen1 be re-ubiquitinated during its hold-over in the TGN? This should provide solid evidence that the TGN is a site of ubiquitination.

2) At present, the manuscript has not excluded a major alternate explanation that the deletion of ROD1 or GGA genes causes a defect in organelle identity resulting in the presence of Sec7 on early endosomes. This would result in an apparent delay in Jen1 traffic at the TGN. One way the authors could exclude this alternate explanation is to examine co-localization of Sec7 and Vps17 in cell lacking ROD1 or GGA genes. If the organelles have maintained their identity, the two markers should not overlap.

3) The data display for Jen1 co-localizing with Sec7 in wt cells is difficult to evaluate. Particularly when one looks at the Figure 4—figure supplement 1, it appears that the two proteins tend to be adjacent rather than co-localized and that the number of co-localized events is small. Without solid co-localization data for Sec7 and Jen1 in wild-type cells, the data presented does not provide a strong enough support for the conclusion that Rod1 normally acts to move Jen1 from the TGN to the vacuole.

Instead of reporting # of structures containing both Jen1 and Sec7, the authors should examine what fraction of Sec7 puncta become Jen1 positive at some point during their lifespan and vice versa. In terms of displaying the images, the current display of 5,10 and 15 minutes does not provide enough time resolution and, while beautiful, the movies are more difficult to analyze than still images. Chymographs or montages of small regions following individual events with greater time resolution would be easier.

4) The authors mention that Jen1 accumulates at the TGN (Figure 3). This could also be due to slower uptake of Jen1 in the *rod1Δ* mutant as they show in Figure 1. However, one would expect an accumulation of Jen1 at the TGN in the *gga1Δ gga2Δ* (Figure 1). Can the authors provide a quantification of Figure 1 comparable to 1A,B? For example, they should perform a time-course looking at the co-localization of Jen1 with *vps17* and *sec7* in the *gga1/2* mutant similar to 3A/B. Along these lines, in Figure 3 Jen1 localizes to the vacuole after 30' Glc in the wt but not in *vps52Δ* or *ypt6Δ*. The authors should provide CMAC staining in all these strains to strengthen this point, since the vacuole could be fragmented in the *vps52Δ*. Since Jen1 is not able to localize to the TGN in *vps52Δ* or *ypt6Δ*, what is the identity of these dots? The authors should address this by colocalization.

[Editors' note: further revisions were requested prior to acceptance, as described below.]

Thank you for resubmitting your work entitled “Integrated control of transporter endocytosis and recycling by the arrestin-related protein Rod1 and Rsp5” for further consideration at *eLife*. Your revised article has been favorably evaluated by Randy Schekman (Senior editor) and a member of the Board of Reviewing Editors. The manuscript has been improved but there are some remaining issues that need to be addressed before acceptance, as outlined below: The data in Figure 5 should be improved. This data was included to address the major hypothesis that a second round of ubiquitination can occur at the TGN. Since this is an important panel the data should be very strong.

The final three lanes show that *ypt6* is epistatic to *gga1* and *gga2* in terms of de-ubiquitination of Jen1 at the 30 minute time point. For some reason (possibly changes in cell physiology due to the triple mutant), Jen1 is expressed at very low levels in these cells. The authors conclude that in this mutant, Jen1 is deubiquitinated whereas in the *gga1,2* double mutant Jen1 is not deubiquitinated. However, there appears to be a slight de-ubiqutination in the *gga1,2* double mutant. Given the vast different levels of protein expression, and use of the highly non-linear HRP detection method for western blots; this needs to be reconfirmed. It is suggested that the authors dilute the double mutant samples such that the 4h lactate signal of the two strains are equal and then analyzed the ubiquitination level of Jen1.

---

## [Author Response]

*1) The authors speculate that Jen1 gets actively de-ubiquitylated after internalization and therefore requires another round of ubiquitylation in the TGN in order to be directed to the vacuole. As this two-step model is a central point of this manuscript they should test what happens if the ubiquitination cycle is blocked and whether in that case a localization of Jen1 to the TGN and Rod1 is still required. Or does aberrant ubiquitination bypass the TGN and/or the requirement for Rod1/Rsp5 on the way to the vacuole? The authors should look at the ubiquitination state in the GGA mutant. Would Jen1 be re-ubiquitinated during its hold-over in the TGN? This should provide solid evidence that the TGN is a site of ubiquitination*.

First, we attempted to alter Jen1’s ubiquitylation cycle as suggested. It should be noted that the Jen1-KR mutant (Figure 4—figure supplement 1, Figure 4—figure supplement 2, Figure 4—figure supplement 3 and Figure 4—figure supplement 4 of the original manuscript) couldn’t be used to answer this question, because although it is expressed and targeted to the plasma membrane, it is not internalized in response to glucose due to its lack of ubiquitylation (see new Figure 2). Thus, we have constructed an in-frame fusion of Jen1 with a non-cleavable (G76V mutant) ubiquitin moiety, under the control of an inducible promoter, in order to have a constitutively ubiquitylated Jen1 construct to monitor whether TGN trafficking and Rod1 are now dispensable. This proved to be tricky because of the toxicity of Jen1- containing constructs in bacteria (yeast shuttle vectors containing WT Jen1 considerably slow down bacteria growth, and the Jen1 - KR presented in the manuscript had to be constructed in yeast because it cannot be amplified in bacteria). We obtained only a few clones that were not rearranged. However, upon their transformation into yeast, we could not detect the expression of the construct, although the sequence was correct. Therefore, and unfortunately, we could not answer this question by this mean in the time allotted.

However, as proposed above, we have looked at Jen1 ubiquitylation status in the *gga1*Δ*gga2*Δ mutant (see new Figure 5—figure supplement 1 and new Figure 5). This revealed that Jen1 indeed accumulates in an ubiquitylated state in this strain suggesting that retaining Jen1 at the TGN leads to its massive ubiquitylation. Furthermore, through the use of a new strain bearing a triple *gga1*Δ*gga2*Δ *ypt6*Δ deletion, we also demonstrate that this increased ubiquitylation depends on endosome-to-TGN trafficking (new Figure 5). This shows that Jen1 ubiquitylation occurs after retrograde trafficking and further supports our model that the TGN is a site of ubiquitylation.

*2) At present, the manuscript has not excluded a major alternate explanation that the deletion of ROD1 or GGA genes causes a defect in organelle identity resulting in the presence of Sec7 on early endosomes. This would result in an apparent delay in Jen1 traffic at the TGN. One way the authors could exclude this alternate explanation is to examine co-localization of Sec7 and Vps17 in cell lacking ROD1 or GGA genes. If the organelles have maintained their identity, the two markers should not overlap*.

Indeed, this explanation had not been addressed in the original manuscript. As suggested, we generated new strains and looked for a potential co-localization of Sec7 and Vps17 in the *rod1* Δ and the *gga1Δgga2Δ* mutant strains. As shown in the new Figure 4—figure supplement 6 and Figure 4—figure supplement 7, in these mutant strains, Sec7 and Vps17 appear in distinct compartments that are reminiscent of those observed in the WT strain (cf. Figure 4—figure supplement 1). Therefore, this hypothesis is unlikely to explain the transient accumulation of Jen1 at a TGN - positive compartment in these strains.

*3) The data display for Jen1 co-localizing with Sec7 in wt cells is difficult to evaluate. Particularly when one looks at the*
Figure 3
*supplement, it appears that the two proteins tend to be adjacent rather than co-localized and that the number of co-localized events is small. Without solid co-localization data for Sec7 and Jen1 in wild-type cells, the data presented does not provide a strong enough support for the conclusion that Rod1 normally acts to move Jen1 from the TGN to the vacuole*.

The TGN is a highly dynamic organelle in yeast. Previous studies have described a LatA-insensitive movement with a velocity of about 100 nm/sec ([34]. BBA 2012). We hypothesized that the slight shift between Jen1 and Sec7 mentioned by the Reviewer was due to the time needed between the acquisitions in each channel (exposure/acquisition + changing channel). Therefore, we used a spinning-disc confocal microscope equipped with two cameras, one for each channel, allowing the simultaneous imaging of the green and red signals (Revolution xD TuCam system, Andor), coupled to the microfluidics setup. As shown in the new Figure 4—figure supplement 3, the co- localization between Sec7 and Jen1 now appears very clearly, indicating that the slight shift is likely an artifact due to the fast mobility of TGN structures. This has been discussed in the manuscript, in the relevant section.

*Instead of reporting # of structures containing both Jen1 and Sec7, the authors should examine what fraction of Sec7 puncta become Jen1 positive at some point during their lifespan and vice versa*.

The data presented in the original version of the manuscript indeed represented the number of Jen1 puncta that are Sec7-positive. As requested, we added an additional graph representing the number of Sec7-positive compartments that are also Jen1-positive in both the WT and *rod1*Δ strains (Figure 4—figure supplement 5).

*In terms of displaying the images, the current display of 5,10 and 15 minutes does not provide enough time resolution and, while beautiful, the movies are more difficult to analyze than still images. Chymographs or montages of small regions following individual events with greater time resolution would be easier*.

Regarding kymographs, they are indeed often used to represent the dynamic behavior of proteins at endocytic sites, for instance, but this is only possible because they are rather static and fixed in space. Unfortunately, the fast mobility of TGN structures in the cell, mentioned above, has precluded us from doing kymographs because we cannot follow with certainty a single TGN structure. In our study, we use endogenously tagged proteins, some of which have a low expression level and/or are tagged with mCherry, which is less photostable than GFP. To be able to image cells for 30 min without too much bleaching, we chose a temporal resolution of 1 image/30 sec, and looked only at one focal plane. In these conditions, TGN structures constantly go in or out of focus, or completely change of localization within the cell between each image, making it impossible to track individual events. This is why we chose an alternative approach, in which we instead studied and quantified populations of puncta for co- labeling with Vps17 or Sec7 and Jen1, and that we think better reflects the situation, although in a less graphic manner.

*4) The authors mention that Jen1 accumulates at the TGN (*Figure 3*). This could also be due to slower uptake of Jen1 in the* rod1Δ *mutant as they show in*
Figure 1*. However, one would expect an accumulation of Jen1 at the TGN in the* gga1Δ gga2Δ *(*Figure 1*). Can the authors provide a quantification of*
Figure 1
*comparable to 1A,B? For example, they should perform a time-course looking at the co-localization of Jen1 with* vps17 *and* sec7 *in the* gga1/2 *mutant similar to 3A/B*.

As proposed, we have now included a quantification of the co-localization between Jen1 and the most relevant of these two markers, i.e. Sec7, in the *gga1*Δ*gga2*Δ mutant vs. WT cells (now displayed in Figure 4). This confirms that Jen1 accumulates at the TGN in the *gga1Δgga2Δ* mutant.

*Along these lines, in*
Figure 3
*Jen1 localizes to the vacuole after 30' Glc in the wt but not in* vps52Δ *or* ypt6Δ*. The authors should provide CMAC staining in all these strains to strengthen this point, since the vacuole could be fragmented in the* vps52Δ.

We performed the experiment again using CMAC staining (new Figure 4). This shows that the Jen1-positive structures that we observed are not fragmented vacuoles and therefore, that Jen1 trafficking to the vacuole is indeed impaired in the mutants studied.

In addition, and because the involvement of the retrograde pathway in post-endocytic sorting was rather unexpected, we now provide additional data in the same panel that strengthen this conclusion. Indeed, we generated new strains deleted for *RGP1* or *RIC1*, encoding components of the Ypt6 GEF complex (Siniossoglou et al. EMBO J, 2000). We observed that these mutants display a similar defect in Jen1 trafficking to the vacuole as that observed in the *ypt6*Δ strain. We also provide data on the loss of Jen1 degradation in the *ypt6*Δ mutant*,* in addition to that in the *vps52Δ* mutant already present in the original version of the manuscript (new Figure 4). These data strengthen the point that retrograde trafficking is required for Jen1 degradation in the vacuole after endocytosis.

*Since Jen1 is not able to localize to the TGN in* vps52Δ *or* ypt6Δ*, what is the identity of these dots? The authors should address this by colocalization*.

Because those are likely early endosome-to-TGN intermediates, we have tried to co-localize these structures with Vps17 and generated the corresponding strain. We obtained only a partial co-localization, which could indicate that they are endosome-derived structures, or instead that we are looking at a fraction of Jen1 that is still transiting at the early endosome. Therefore, it is for the moment hard to conclude on the identity of these structures. Besides, although this is an interesting question, we feel that this is more related to the precise characterization of the *ypt6*Δ or *vps52Δ* mutants, than to our study on the place of action of Rod1 in endocytosis, and may require other guesses and more experiments.

[Editors' note: further revisions were requested prior to acceptance, as described below.]

*The data in*
Figure 5
*should be improved. This data was included to address the major hypothesis that a second round of ubiquitination can occur at the TGN. Since this is an important panel the data should be very strong*.

*The final three lanes show that ypt6 is epistatic to* gga1 *and* gga2 *in terms of de-ubiquitination of Jen1 at the 30 minute time point. For some reason (possibly changes in cell physiology due to the triple mutant), Jen1 is expressed at very low levels in these cells. The authors conclude that in this mutant, Jen1 is deubiquitinated whereas in the* gga1,2 *double mutant Jen1 is not deubiquitinated. However, there appears to be a slight de-ubiqutination in the* gga1,2 *double mutant. Given the vast different levels of protein expression, and use of the highly non-linear HRP detection method for western blots; this needs to be reconfirmed. It is suggested that the authors dilute the double mutant samples such that the 4h lactate signal of the two strains are equal and then analyzed the ubiquitination level of Jen1*.

Thank you for this comment on our blot, which is indeed an important panel. The experiment had to be performed again, because the ubiquitylation pattern usually appears diffuse after a few weeks of storage at -20°C. As per the reviewer’s request, we have changed the amount loaded to be able to compare Jen1 ubiquitylation in the three strains on the same exposure. The results, now provided in new Figure 5, confirm our previous data and are in line with our conclusions. A full blot is also shown below (Figure 9).Author response image 1.

In the figure legend, we added the following comment:

“Note that Jen1-GFP is expressed at a lower level in the triple *ypt6Δ gga1Δ gga2Δ* mutant, therefore samples were loaded so that a comparable signal is observed in each strain.”